# ADAPTIVE MAMBA NEURAL OPERATORS

**Zeyuan Song & Zheyu Jiang**
School of Chemical Engineering
Oklahoma State University
Stillwater, OK 74078, USA
{taekwon.song,zheyu.jiang}@okstate.edu

## ABSTRACT

Accurately solving partial differential equations (PDEs) on arbitrary geometries and a variety of meshes is an important task in science and engineering applications. In this paper, we propose Adaptive Mamba Neural Operators (AMO), which integrates reproducing kernels for state-space models (SSMs) rather than the kernel integral formulation of SSMs. This is achieved by constructing Takenaka-Malmquist systems for the PDEs. AMO offers new representations that align well with the adaptive Fourier decomposition (AFD) theory and can approximate the solution manifold of PDEs on a wide range of geometries and meshes. In several challenging benchmark PDE problems in the fields of fluid physics, solid physics, and finance on point clouds, structured meshes, regular grids, and irregular domains, AMO consistently outperforms state-of-the-art solvers in terms of relative $L^2$ error. Overall, this work presents a new paradigm for designing explainable neural operator frameworks. The code is available at https://github.com/checlams/AMO.

## 1 INTRODUCTION

A wide range of scientific and engineering phenomena, including fluid dynamics, heat and mass transport, structural mechanics, and cell growth, can be characterized and modeled by partial differential equations (PDEs). Most nonlinear PDEs do not have analytical solutions and need to be solved numerically. Traditional discretization-based approaches for solving PDEs can be computationally expensive. To speed up the solution process, neural operators have recently been proposed as an extension of neural networks to learn the infinite-dimensional solution operators of various PDE problems. It has been proven that, with finite-dimensional solutions as training data, neural operators can accurately learn the infinite-dimensional solution space. Once learned, neural operators are mesh-independent, so neural operators trained on coarse grids can generalize to finer grids.

Frequency-based neural operators, such as Fourier neural operator (FNO) (Li et al., 2020), wavelet neural operator (WNO) (Tripura & Chakraborty, 2023), multiwavelet transform (MWT) (Gupta et al., 2021), U-shaped neural operator (Rahman et al., 2022), spectral neural operator (Fanaskov & Oseledets, 2023), and latent spectral model (LSM) (Wu et al., 2023), are attractive since the solution space of many PDEs can be naturally expressed in spectral bases. Frequency-based neural operators approximate the PDE solutions by learning how frequencies evolve, and nonlinear terms become convolution in the associated frequency domain. However, the performance of existing frequency-based neural operators may deteriorate in irregular geometries (Li et al., 2023), as their associated bases could lose orthogonality and eigenfunction properties in irregular domains (Lingsch et al., 2023; Chen et al., 2024). As a result, retaining these important properties for kernels and bases for irregular domains is critical.

Along this line, a recently proposed neural operator solver, latent Mamba operator (LaMO) (Tiwari et al., 2025), shows great promise in capturing PDE solutions on irregular domains. LaMO integrates the efficiency of state-space models (SSMs) in latent space with the expressive power of kernel integral formulations in neural operators. Although the selective convolution kernels utilized in LaMO can effectively capture PDE solutions on irregular domains, their lack of orthogonality property may lead to spectral mixing. Furthermore, the kernels in LaMO are finite-order linear dynamic filters (Gu & Dao, 2023), which may introduce a low-pass filtering bias, leading to poor re-

covery of high-frequency and singular features (Gu et al., 2021; Gu & Dao, 2023). In the illustrative experiments discussed in Appendix B, we show that LaMO suffers from deviation in the propagation of high-frequency perturbations for 1-D advection PDE, and it fails to capture the singularities in 2-D Darcy flow equation with fractal noise as the permeability field.

Recognizing the fact that LaMO lacks frequency-domain implementation, here we propose Adaptive Mamba Neural Operator (AMO), a novel neural operator architecture that synergizes an adaptive Fourier decomposition with the efficiency of structured SSMs in the frenquency domain (Gu & Dao, 2023; Parnichkun et al., 2024). AMO parameterizes the SSM transfer function in a Takenaka-Malmquist (TM) system in a reproducing kernel Hilbert space (RKHS), thus allowing state-free kernel construction and inference directly on the spectrum. The Mamba blocks in AMO serve as rational filters while retaining linear-time selective scanning. Furthermore, it turns out that AMO structure resembles adaptive Fourier decomposition (AFD), a novel signal decomposition technique achieving higher accuracy and significant computational speedup compared to conventional signal decomposition methods (Qian, 2010; Qian et al., 2012). The architecture and design of AMO is fully guided by the AFD theory, thereby improving the mathematical explainability and groundness of AMO,

Overall, our key contributions are summarized as follows:

1. AMO is the first neural operator which explicitly incorporates TM systems and Fourier-based methods into the Mamba structure. AMO accurately solves the PDE problems on diverse geometries and effectively handles singularities and long-range dependencies of PDE solutions, Furthermore, we develop theoretical foundations for AMO and prove that AMO performs AFD approximation of PDE solutions.

2. The design of every component of AMO is fully guided by the AFD theory, leading to a mathematically interpretable and grounded architecture. Using a TM layer, AMO projects the input into TM systems in a Hardy space, and constructs the reproducing kernels from adaptively selected poles. These adaptive poles serve to construct the reproducing kernels adaptively. We demonstrate the importance of utilizing adaptive poles as opposed to fixed poles and investigate how the number of adaptive poles influences the performance of AMO.

3. AMO outperforms state-of-the-art neural operator solvers in terms of accuracy across a diverse set of benchmark PDE problems, including plasticity, elasticity, airfoil, pipe flow, Navier-Stokes, and Darcy flow on various geometries. It also achieves outstanding performance in financial applications, such as solving the Black-Scholes equation for the European option pricing problem.

## 2 RELATED WORK

**Frequency-based neural operators.** Early advancements in operator learning exploited spectral decompositions to encode global information efficiently. A notable example is FNO (Li et al., 2020), which parameterizes integral kernels in the Fourier domain to enable resolution-invariance. However, FNO does not generalize well to irregular geometries (Li et al., 2020). Later, Geo-FNO (Li et al., 2023) was proposed to solve PDEs on general geometries. U-FNO (Wen et al., 2022) introduced architectural modifications to better capture localized details while maintaining FNO's global properties. Meanwhile, F-FNO (Tran et al., 2021) generalizes the FNO architecture for more efficient spectral layers and deeper architectures. On the other hand, neural operators based on the wavelet transform include WNO (Tripura & Chakraborty, 2023), MWT (Gupta et al., 2021), Padé (Gupta et al., 2022), and CMWNO (Xiao et al., 2023a). Fourier and wavelet transforms are both special cases of spectral decomposition, and neural operators based on spectral decomposition has recently been proposed (Fanaskov & Oseledets, 2023).

**Attention-based neural operators.** Attention mechanisms have been widely studied in neural operator domain. Some of the notable works include orthogonal attention (Xiao et al., 2023b), physics-cross-attention (Wang & Wang, 2024), and nonlocal attention (Yu et al., 2024). The Transformer structure is also a promising building block for neural operators. Some of the related works include OFormer (Li et al., 2022), LSM (Wu et al., 2023), and Transolver (Wu et al., 2024). However, Transformers struggle to capture kernel integral transforms efficiently in complex, high-dimensional continuous PDEs (Guibas et al., 2021).

**SSM-based neural operators.** To address the computational inefficiency of Transformer-based neural operators, SSM and Mamba emerge as promising architectures for neural operator designs (Tiwari et al., 2025). Previous studies of SSM-based neural operators (Zheng et al., 2024; Cheng et al., 2024; Hu et al., 2024; Tiwari et al., 2025) have been applied to nonlinear PDEs on irregular geometries and dynamical systems. These works incorporate traditional SSMs with different scan strategies without considering the information in the frequency domain. On the other hand, our AMO considers the frequency information via its explicit kernel and SSMs from a transfer function perspective (Parnichkun et al., 2024).

## 3 ADAPTIVE FOURIER MAMBA OPERATOR

### 3.1 PROBLEM STATEMENT

We frame our task as learning a solution operator for a family of parametric PDEs. In general, consider a PDE defined on a spatial domain $\Omega \subset \mathbb{R}^d$ and a time interval $(0, T]$:

$$\mathcal{L}_a[u(x,t)] = f(x,t), \quad \forall(x,t) \in D \times (0,T], \tag{1}$$

which is subject to a set of initial and boundary conditions. Here, the parameter function $a \in \mathcal{A}$ specifies the coefficients and initial and boundary conditions of Equation 1. In operator learning, our goal is to construct an accurate approximation for $\mathcal{G} : \mathcal{A} \to \mathcal{F}(D \times [0,T])$, which maps the parameter function $a$ to the corresponding solution function $u(x,t) \in \mathcal{F}$, via a parametric mapping $\mathcal{G}_\theta$. The aim is to learn $\theta$ such that $\mathcal{G}_\theta \approx \mathcal{G}$ from a set of training data $\{(a_j, u_j)\}_j$.

### 3.2 AMO ARCHITECTURE

AMO is a novel neural operator architecture that synergizes the mathematical groundness of AFD theory with the efficiency of structured SSMs in the frenquency domain (Gu & Dao, 2023; Parnichkun et al., 2024). Different from LaMO (Tiwari et al., 2025), which compresses the physical tokens into a fixed-size latent representation, AMO utilizes a multi-layer fully-connected feedforward neural network (MLP) to first map the encoded tokens to their counterparts on the reproducing kernel Hilbert space (RKHS), and then iteratively refine them by a series of processing blocks. Each block uniquely integrates two components: (i) a TM layer containing global spectral transform via data-dependent TM bases, and (ii) a bidirectional SSM (Gu et al., 2021; Gu & Dao, 2023) parameterized by transfer functions in the frequency domain (Parnichkun et al., 2024) to efficiently capture long-range dependencies within the RKHS.

**Neural architecture.** Given the parameter function (input) $a$, the output of AMO, denoted as $\hat{u}_{N,\theta}$, is:

$$\hat{u}_{N,\theta} = \mathcal{G}_\theta(a) = \left(\mathcal{Q} \circ \mathcal{S}^N \circ \mathcal{L}^N \circ \cdots \circ \mathcal{S}^1 \circ \mathcal{L}^1 \circ \mathcal{R} \circ \mathcal{P}\right)(a), \tag{2}$$

where $\circ$ is the function composition, $N$ is the number of processing blocks, $\mathcal{P}$ is the lifting operator which encodes into a lower-dimensional space (maps the input to the first latent representation $\mathbf{z}_0$) (Tiwari et al., 2025; Li et al., 2020), $\mathcal{Q}$ is the corresponding projection operator mapping the lower-dimensional space back to the original space (maps the final latent representation $\mathbf{z}_{N+1}$ to the output) (Tiwari et al., 2025; Li et al., 2020), $\mathcal{R}$ is a multi-layer neural network mapping the physical token to an RKHS, $\mathcal{L}^i = \text{SSM}^i \circ \text{TM}^i$ $(i = 1, \ldots, N)$ is the processing block of AMO (which consists of a TM layer and a bidirectional SSM), and $\mathcal{S}^i$ $(i = 1, \ldots, N)$ are aggregation layers with skip connections. These aggregation layers not only receive the final output from the layer sequence but also have access to the intermediate outputs from each of the preceding layers.

**The lifting operator,** $\mathcal{P}$, projects the $N_s$ physical token inputs into a compressed set of $M$ encoded tokens, where $M \ll N_s$. This projection is achieved via a cross-attention mechanism. A learnable query array, $\mathbf{L} \in \mathbb{R}^{M \times D_{\text{embed}}}$, acts as the query. The key and value pairs are constructed by combining a linear projection of the input features $\mathbf{x}_{\text{phys}}$ with a positional embedding of their coordinates $\mathbf{g}_{\text{phys}}$ generated by a positional encoding network PEN. Here, $\mathbf{x}_{\text{phys}} \in \mathbb{R}^{N_s \times D_{\text{in}}}$ stacks the feature vectors $\{\mathbf{x}_i\}_{i=1}^{N_s}$ and $\mathbf{g}_{\text{phys}} \in \mathbb{R}^{N_s \times d}$ stacks the coordinates $\{\mathbf{g}_i\}_{i=1}^{N_s}$, and the physical token is essentially

pair $(\mathbf{g}_i, \mathbf{x}_i)$. The process for generating the initial representation $\mathbf{z}_0$ is formally defined as:

$$
\begin{aligned}
\mathbf{kv} &= \mathrm{Linear}(\mathbf{x}_{\mathrm{phys}}) + \mathrm{PEN}(\mathbf{g}_{\mathrm{phys}}), \\
\mathbf{z}_0' &= \mathrm{CrossAttn}(\mathrm{query} = \mathbf{L}, \mathrm{key} = \mathbf{kv}, \mathrm{value} = \mathbf{kv}), \\
\mathbf{z}_0 &= \mathbf{z}_0' + \mathrm{FFN}(\mathbf{z}_0'),
\end{aligned}
\tag{3}
$$

where the output of the cross-attention module is processed through a residual connection and a standard feed-forward network FFN.

**The mapping operator,** denoted by $\mathcal{R}$, acts on the encoded representation produced by the lifting operator $\mathcal{P}$, which transforms this discrete encoded tokens into a representation within a continuous function space. Let $\mathbf{z}_0 \in \mathbb{R}^{M \times D_{\mathrm{embed}}}$ be the set of encoded tokens generated by $\mathcal{P}$, the operator $\mathcal{R} : \mathbb{R}^{M \times D_{\mathrm{embed}}} \to \mathcal{H}$ maps this representation to its counterpart in an RKHS $\mathcal{H}$. This mapping is typically implemented as a multi-layer fully-connected feedforward network MLP, which processes each token independently as:

$$
\mathbf{z}_1 = \mathcal{R}(\mathbf{z}_0) = \mathrm{MLP}(\mathbf{z}_0),
\tag{4}
$$

where $\mathbf{z}_1$ denotes the projected tokens in the RKHS. We remark that, the mapping operator $\mathcal{R}$ maps the encoded tokens $\mathbf{z}_0$ to the new tokens $\mathbf{z}_1$ in $\mathcal{H}$ without knowing the physical information $\mathbf{x}_{\mathrm{phys}}$ and $\mathbf{g}_{\mathrm{phys}}$.

**The TM layer,** denoted by $\mathrm{TM}^i$ ($i = 1, \ldots, N$), performs a global convolution via a spectral transform, where the reproducing kernels and TM bases are constructed from data-dependent poles. To define the reproducing kernels, we parameterize a small MLP to predict a set of $i$ complex values called "poles" $\{a_k\}_{k=1}^i$ (denoted as $a_{1:i}$) located in the unit disk $\mathbb{D} = \{z \in \mathbb{C} : |z| < 1\}$ from tokens $\mathbf{z}_i$. Once we have the set of poles, we can explicitly define the reproducing kernel $K_a(z)$ as:

$$
K_a(z) = \frac{1}{1 - \overline{a}z},
\tag{5}
$$

where $z \in \mathcal{H}$ and $a$ is a single pole satisfying $|a| < 1$. Intuitively, we remark that each pole can be viewed as a "tuning knob" that selects a particular spatial pattern in the solution, with its location in the complex plane controlling how localized that pattern is. Adaptive poles allow AMO to survey more heavily in regions where the parameters change rapidly, while using fewer poles in smooth regions. Across layers, the poles evolve from broad, coarse patterns in early layers to more refined, problem-specific patterns in deeper layers.

To generalize on irregular geometries, the kernels in Equation 5 need to be modified to become orthonormal. These modified kernels are also known as the TM bases due to their deep connection to TM systems. The first basis, denoted as $\mathscr{B}_1$, is simply the normalized kernel of Equation 5 with pole $a_1$ as $\mathscr{B}_1(z; a_1) = \frac{\sqrt{1 - |a_1|^2}}{1 - \overline{a_1}z}$. Then, we start with $\frac{\sqrt{1 - |a_2|^2}}{1 - \overline{a_2}z}$, but it is not orthogonal to $\mathscr{B}_1$. We reach the orthogonality by subtracting its projection onto $\mathscr{B}_1$, and we get $\mathscr{B}_2(z; a_{1:2}) = \frac{\sqrt{1 - |a_2|^2}}{1 - \overline{a_2}z} \left( \frac{z - a_1}{1 - \overline{a_1}z} \right)$ after normalization. This way, the bases $\mathscr{B}_i$ are finally formulated as:

$$
\mathscr{B}_i(z; a_{1:i}) = \frac{\sqrt{1 - |a_i|^2}}{1 - \overline{a_i}z} \prod_{j=1}^{i-1} \frac{z - a_j}{1 - \overline{a_j}z},
\tag{6}
$$

where $z \in \mathcal{H}$ and $a_{1:i}$ are poles learned by the small MLP satisfying $|a_k| < 1$ for $k = 1, \ldots, i$. Overall, the $i$-th TM layer $\mathrm{TM}^i$ applies a small MLP $\mathbf{z_i} \mapsto a_{1:i}$, and then construct the TM bases $\mathscr{B}_i$ according to 6. We remark that, the tokens $\mathbf{z}_i$ will be kept as the input of $\mathrm{SSM}^i$ along with the TM bases $\mathscr{B}_i$.

**Bidirectional SSM block** is effective in solving PDEs on irregular geometries (Tiwari et al., 2025) and employs inherent kernel integrals. However, this inherent kernel does not contain information in the frequency domain, thereby falling short in capturing high-frequency and singular features. To address this limitation, we utilize the transfer function in training SSMs in the frequency domain (Parnichkun et al., 2024). The SSM block $\mathrm{SSM}^i$ generates the spectrum of output in the frequency domain $Y_i(e^{i\omega})$ as the product of the spectrum of input $Z(e^{i\omega})$ and the transfer function $H_i(e^{i\omega})$,

i.e., $Z(e^{i\omega})H_i(e^{i\omega})$. We point out that the output is essentially the coefficient of discrete AFD operation with the form $\langle \mathbf{z_i}, \mathscr{B}_i \rangle$ (Qian, 2010; Qian et al., 2011), where the inner product is defined as $\langle x, f \rangle = \frac{1}{\tilde{N}} \sum_{n=0}^{\tilde{N}-1} x[n]\overline{f(e^{i2\pi n/\tilde{N}})}$. Here, $\tilde{N}$ denotes the length of signal $x = \{x[n]\}_{n=0}^{\tilde{N}-1}$.

Let us consider the impulse response $h_i$ of SSM block $\text{SSM}^i$ (linear time-invariant system) as:

$$h_i[n] = \frac{1}{2\pi} \int_0^{2\pi} \overline{\mathscr{B}_i\left(e^{i\omega}; a_{1:i}\right)} e^{i\omega n} \, d\omega. \tag{7}$$

Then, the corresponding transfer function $H_i$ can be obtained as:

$$H_i(e^{i\omega}) = \overline{\mathscr{B}_i\left(e^{i\omega}; a_{1:i}\right)}. \tag{8}$$

By setting the transfer function of SSM to be Equation 8, the SSM block computes a correlation of the input $\mathbf{z}_i$ and $\mathscr{B}_i$:

$$Y_i(e^{i\omega}) = H_i(e^{i\omega})X(e^{i\omega}) = \overline{\mathscr{B}_i(e^{i\omega}; a_{1:i})} \, X(e^{i\omega}) \tag{9}$$

in the frequency domain. In the time domain, Equation 9 leads to the update of $\mathbf{z_i}$:

$$\hat{\mathbf{z}}_{\mathbf{i+1}}[\ell] = (h_i * \mathbf{z_i})[\ell] = \sum_{n=0}^{M-1} \mathbf{z_i}[n]\overline{\mathscr{B}_i\left(e^{i2\pi(n-\ell)/M}; a_{1:i}\right)}, \tag{10}$$

where $\ell$ denotes the time shift in the correlation operations. The zero-lag sample gives the final output:

$$\hat{\mathbf{z}}_{\mathbf{i+1}}[0] = (h_i * \mathbf{z_i})[0] = \sum_{n=0}^{M-1} \mathbf{z_i}[n]\overline{\mathscr{B}_i\left(e^{i2\pi n/M}; a_{1:i}\right)} = \langle \mathbf{z_i}, \mathscr{B}_i \rangle. \tag{11}$$

**Aggregation layers** $\mathcal{S}^i$ has $N$ neural layers and combines the skip connection $\mathbf{z_i}$ with the intermediate outputs $\hat{\mathbf{z}}_{\mathbf{i+1}}[0] = \mathcal{L}^i(\mathbf{z_i})$ and $\mathscr{B}_i = \text{TM}^i(\mathbf{z_i})$:

$$\begin{aligned}
\mathbf{z_2} &= \mathcal{S}^i(\mathbf{z_1}, \hat{\mathbf{z}}_{\mathbf{2}}[0], \mathscr{B}_1) = \hat{\mathbf{z}}_{\mathbf{2}}[0] \odot \mathscr{B}_1 \quad \text{for } i = 1, \\
\mathbf{z_{i+1}} &= \mathcal{S}^i(\mathbf{z_i}, \hat{\mathbf{z}}_{\mathbf{i+1}}[0], \mathscr{B}_i) = \mathbf{z_i} + (\hat{\mathbf{z}}_{\mathbf{i+1}}[0] \odot \mathscr{B}_i) \quad \text{for } i > 1,
\end{aligned} \tag{12}$$

where $\odot$ denotes the element-wise (Hadamard) product.

**Output.** Finally, the output of $\hat{u}_{N,\theta}$ is the projection of $\mathbf{z_{N+1}}$ by the local transformation $\mathcal{Q}$ as (Li et al., 2020):

$$\hat{u}_{N,\theta} = \mathcal{Q}\left(\sum_{i=1}^{N+1}\left(\sum_{n=0}^{M-1} \mathbf{z_i}[n]\overline{\mathscr{B}_i\left(e^{i2\pi n/M}; a_{1:i}\right)}\right) \odot \mathscr{B}_i\right). \tag{13}$$

## 4  PROPERTIES OF AMO

**Connections to AFD theory.** Adaptive Fourier decomposition (AFD) is a novel signal decomposition technique that leverages the Takenaka-Malmquist system and adaptive orthogonal bases (Qian, 2010; Qian et al., 2012). It admits a proved convergence of any signal $s \in \mathcal{H}$ such that $s = \sum_{i=1}^{\infty} \langle s, \mathscr{B}_i \rangle \mathscr{B}_i$ (Qian et al., 2011; Wang et al., 2022) for the chosen orthonormal bases $\mathscr{B}_i$ (Saitoh et al., 2016). Thus, the output of Equation 12 $\mathbf{z_{i+1}}$, is equivalent to the AFD operation, i.e., $\mathbf{z_{i+1}} = \sum_{k=1}^{i} \langle \mathbf{z_k}, \mathscr{B}_k \rangle \mathscr{B}_k$. Furthermore, the output in Equation 13 can be approximated as $\hat{u}_{N,\theta} = \mathcal{Q}\left(\sum_{i=1}^{N+1} \langle \mathbf{z_i}, \mathscr{B}_i \rangle \mathscr{B}_i\right) \approx \sum_{i=1}^{N+1} \langle \hat{u}_{i-1,\theta}, \mathscr{B}_i \rangle \mathscr{B}_i$, where $\hat{u}_{i-1,\theta} = \mathcal{Q}(\mathbf{z_i})$. This is also equivalent to the AFD operation. Thus, several theoretical properties of AMO, including convergence and error bound (see theorems and proofs in Appendix D), can be guaranteed with efficiently large layers, thanks to AMO's deep connections with AFD theory.

**Connections to Parnichkun et al. (2024).**   Parnichkun et al. (2024) proposed a state-free inference of SSMs by learning the coefficients of the rational transfer function $H$ instead of the traditional state-space matrices $A$, $B$, and $C$ (Gu & Dao, 2023), which is called rational transfer function (RTF) approach. Specifically, the RTF learns $H$ as:

$$H(z) = h_0 + \frac{b_1 z^{-1} + b_2 z^{-2} + \cdots + b_n z^{-n}}{1 + a_1 z^{-1} + a_2 z^{-2} + \cdots + a_n z^{-n}}, \tag{14}$$

where $a_i$, $b_i$, and $h_0$ are denominator coefficients, numerator coefficients, and feedthrough term, respectively. When it comes to AMO, we push the formulation of transfer function in Equation 8 and learn the rational transfer function by learning the poles $a_{1:n}$ (for $n$ terms). In Appendix E, we show that our way of learning poles leads to a similar form of Equation 14 with $n$ learned parameters (poles) as opposed to learning $2n + 1$ parameters in RTF.

**Computational complexity.**   In terms of computational complexity, AMO has an overall computational complexity of $\mathcal{O}\big(N(M \log M + MD)\big) + \mathcal{O}(N_s MD)$. The former is from the processing block, whereas the latter comes from $\mathcal{P}$ and $\mathcal{Q}$. When $M$ is treated as a constant with $M \ll N_s$ and a local decoder is used, the dominant cost reduces to $\mathcal{O}(N_s D) + \mathcal{O}(N M \log M)$. Consequently, the complexity grows linearly with the number of mesh points $N_s$. With mesh size fixed, it is approximately linear in the number of latent tokens $M$ and the number of blocks $N$.

## 5   Numerical Experiments

To illustrate the effectiveness of AMO, we conduct numerical experiments with multiple baseline neural operators on diverse datasets including three categories: (i) regular grids: 2-D Darcy flow equation and 2-D Navier-Stokes equation (Li et al., 2020), (ii) irregular geometries: plasticity, airfoil, pipe, and elasticity (Li et al., 2023), (iii) PDEs with singularities: European option pricing under the Black-Scholes equation, and 3-D Brusselator (reaction-diffusion) equation from Cao et al. (2024) (see Appendix B).

**Metric.**   In the training and evaluation stage, we utilize relative $L^2$ error as the metric for accuracy for all problems:

$$\text{Rel-}L^2 = \frac{1}{\mathcal{N}} \sum_{i=1}^{\mathcal{N}} \frac{||\mathcal{G}_\theta(a_i) - \mathcal{G}(a_i)||_{L^2}}{||\mathcal{G}(a_i)|_{L^2}}, \tag{15}$$

where $\mathcal{N}$ denotes the number of samples. We also consider training time, the number of parameters, and/or GPU memory usage as metrics for computational efficiency.

**Implementation details.**   For baselines, we follow the implementation settings of their works. Note that the architecture of FNO (Li et al., 2020) has been updated after publication, we evaluate FNO using the newest architecture. For AMO, we train $500$ epochs on all datasets. We use AdamW optimizer with decoupled weight decay $1 \times 10^{-5}$, base learning rate $2 \times 10^{-4}$, and a cosine decay schedule (Loshchilov & Hutter, 2017) with a linear warm-up over the first $10\%$ of total steps. The nonlinearity is GELU inside the processing blocks. We clip global grad-norm at $0.5$ each step. Unless stated otherwise, we use batch size $16$, latent width $128$, $64$ latent tokens, $32$ adaptive poles, and $4$ processing blocks with SSM state size $16$, depthwise 1-D convolution (per channel) of kernel size $4$, channel expansion ratio $2$. Experiments are conducted on a Linux workstation running Ubuntu (kernel 6.14, glibc 2.39) with Python 3.13.5 (Anaconda), PyTorch 2.8.0+cu129 (CUDA 12.9), an AMD Ryzen 9 9950X (16-core) processor, and a single NVIDIA GeForce RTX 4090 (48 GB) GPU. CUDA is enabled.

### 5.1   Numerical results of benchmark datasets

Table 1 shows the comprehensive comparison with various baselines on the six benchmark problems. Among those problems, N-S and Darcy flow datasets apply regular grids, elasticity dataset uses point clouds, whereas others are generated under structured meshes (Li et al., 2020; 2023). AMO consistently outperforms existing SOTA models by an average improvement of $28.42\%$. In particular, for airfoil, Darcy, and N-S datasets, the relative $L^2$ error decreased more than $30\%$ compared to

the existing SOTA models, demonstrating the superior performance of AMO compared to existing frequency-, transformer-, and Mamba-based models when solving complex dynamics and handling irregular geometries. To solve the complex dynamics, Tiwari et al. (2025) incorporates latent representations and SSMs, which can be considered as integral kernels without orthogonality. Meanwhile, ONO (Xiao et al., 2023b) uses an orthogonal attention to ensure orthogonality. Numerical results on irregular geometries, including elasticity ($0.0050 \rightarrow 0.0043$), plasticity ($0.0007 \rightarrow 0.0006$), airfoil ($0.0041 \rightarrow 0.0020$), and pipe ($0.0026 \rightarrow 0.0023$), show that the systematic integration of orthonormal kernels and SSMs leads to an exact AFD approximation and in turn improves PDE solution accuracy in irregular geometries.

Table 1: Relative $L^2$ error comparisons of AMO with baselines across six benchmark datasets. Lower relative $L^2$ error is better. We quantify the improvement as the gain of AMO relative to the $L^2$ error of the second best model. **Bold** means the best model, underline means the second best model, red means the third best model, and blue means the fourth best model.

| Models | Elasticity | Plasticity | Airfoil | Pipe | N-S | Darcy |
|---|---|---|---|---|---|---|
| FNO (Li et al., 2020) | 0.0229 | 0.0074 | 0.0138 | 0.0067 | 0.0417 | 0.0052 |
| U-FNO (Wen et al., 2022) | 0.0239 | 0.0039 | 0.0269 | 0.0056 | 0.2231 | 0.0183 |
| F-FNO (Tran et al., 2021) | 0.0263 | 0.0047 | 0.0078 | 0.0070 | 0.2322 | 0.0077 |
| LNO (Wang & Wang, 2024) | 0.0052 | 0.0029 | 0.0051 | 0.0026 | 0.0845 | 0.0049 |
| ONO (Xiao et al., 2023b) | 0.0118 | 0.0048 | 0.0061 | 0.0052 | 0.1195 | 0.0076 |
| WMT (Gupta et al., 2021) | 0.0359 | 0.0076 | 0.0075 | 0.0077 | 0.1541 | 0.0082 |
| Galerkin (Cao, 2021) | 0.0240 | 0.0120 | 0.0118 | 0.0098 | 0.1401 | 0.0084 |
| LSM (Wu et al., 2023) | 0.0218 | 0.0025 | 0.0059 | 0.0050 | 0.1535 | 0.0065 |
| OFormer (Li et al., 2022) | 0.0183 | 0.0017 | 0.0183 | 0.0168 | 0.1705 | 0.0124 |
| Transolver (Wu et al., 2024) | 0.0062 | 0.0013 | 0.0053 | 0.0047 | 0.0879 | 0.0059 |
| Transolver++ (Luo et al., 2025) | 0.0064 | 0.0014 | 0.0051 | 0.0027 | 0.1010 | 0.0089 |
| LAMO (Tiwari et al., 2025) | 0.0050 | 0.0007 | 0.0041 | 0.0038 | 0.0460 | 0.0039 |
| **AMO (ours)** | **0.0043** | **0.0006** | **0.0020** | **0.0023** | **0.0278** | **0.0021** |
| **Improvement** | 14.0% | 14.3% | 51.2% | 11.5% | 33.3% | 46.2% |

**Computational Efficiency.** To explore the computational efficiency of AMO, we focus on Darcy and airfoil problems. On average, AMO reaches $46.2\%$ and $51.2\%$ reduction in training time over SOTA models in these two problems, as shown in Figure 1. With light architectures and small GPU memory, AMO achieves the best training speed. Compared to the SOTA neural operator, LaMO (Tiwari et al., 2025), AMO is $\sim 1.2\times$ faster and $\sim 2.5\times$ lighter with similar GPU memory. Instead of using orthogonal attention as in ONO (Xiao et al., 2023b), AMO employs bases in the orthogonal form (Equation 6), which does not require an orthogonalization process, thereby saving $\sim 2.7\times$ in training time and $\sim 3\times$ in GPU memory compared to ONO.

**Scalability.** We examine the computational scalability of AMO on 2-D Darcy flow problem. From Table 2, we observe that, as the grid dimension changes from 64 to 128 ($N_s$ becomes 4 times larger), both training and inference times increase approximately linearly (by about 4 times), which aligns with the computational complexity result mentioned earlier. The memory usage remains relatively constant with only a slight increase. This reflects the architectural characteristics of AMO, where the main computations (SSM blocks) are performed on $M$ latent tokens rather than on $N_s$ physical points, and thus the memory footprint is largely decoupled from the input resolution $N_s$.

Table 2: AMO is computationally scalable with respect to input resolution $N_s$.

| Grid dimensions | Grid size $N_s$ | Training time (sec/epoch) | Inference time (sec/epoch) | GPU memory (GB) |
|---|---|---|---|---|
| $64 \times 64$ | 4096 | 14.0 | 0.007 | 2.3 |
| $128 \times 128$ | 16384 | 52.5 | 0.28 | 2.4 |
| $256 \times 256$ | 65536 | 205.0 | 1.12 | 2.7 |

**Learned pole distributions across layers.** To understand how the adaptive poles are selected and evolved, Figures 5 and 6 showcase the distributions per layer for 2-D Darcy flow and 3-D Brusselator equations. The learned poles of AMO on Darcy flow problem tend to approach to the boundary of the unit disk, while those on the Brusselator problem tend to be in the interior of the unit disk. The reason is that, the challenging characteristics and singularities of the Darcy flow problem are located at the boundaries, and then more adaptive poles would be put there. Meanwhile, the complexity of the Brusselator problem does not come from the boundaries. It comes from the local, non-linear reaction that happens at every single point inside the domain. Therefore, most of the learned poles should be put inside the unit disk.

## 5.2 EUROPEAN OPTIONS PRICING

To demonstrate the versatility of AMO in solving different PDEs in different contexts, we consider the European calls/puts problem modeled using the Black–Scholes equation with continuous dividend yield $q$. For contract/market parameters $(r, \sigma, q, K, T, \texttt{is\_call})$, the price $V(S,t)$ satisfies the Black–Scholes equation (Barles & Soner, 1998):

$$\partial_t V + \tfrac{1}{2}\sigma^2 S^2 \partial_{SS} V + (r - q)S \partial_S V - rV = 0, \quad S \in [S_{\min}, S_{\max}], \ t \in [0, T], \quad (16)$$

with terminal payoff $V(S, T) = \max(\pm(S - K), 0)$ (+ sign for calls, − for puts) and the linear boundary conditions $V(0, t) = 0$ for calls, $V(0, t) = Ke^{-r(T-t)}$ for puts, and controlled growth as $S \to \infty$. This problem setting leads to two singular features: (i) the terminal payoff kink at $S = K$ (jump in $\partial_S V$, concentration in $\partial_{SS} V$) as $t_{\text{norm}} \uparrow 1$; and (ii) degeneracy near small $S$ as a result of the $S^2 \partial_{SS} V$ diffusion term. Our goal is to learn the operator that maps the parameters $(r, \sigma, q, K, T, \texttt{is\_call})$ to the price $V(S, t)$. By comparing AMO with a set of top-performing solvers, we observe from Table 3 that average improvements of 25%, 4.1%, and 52.7% have been achieved by AMO in terms of relative $L^2$ error, training time, and parameter counts, respectively. This indicates that AMO can accurately and efficiently solve PDE problems with singular features.

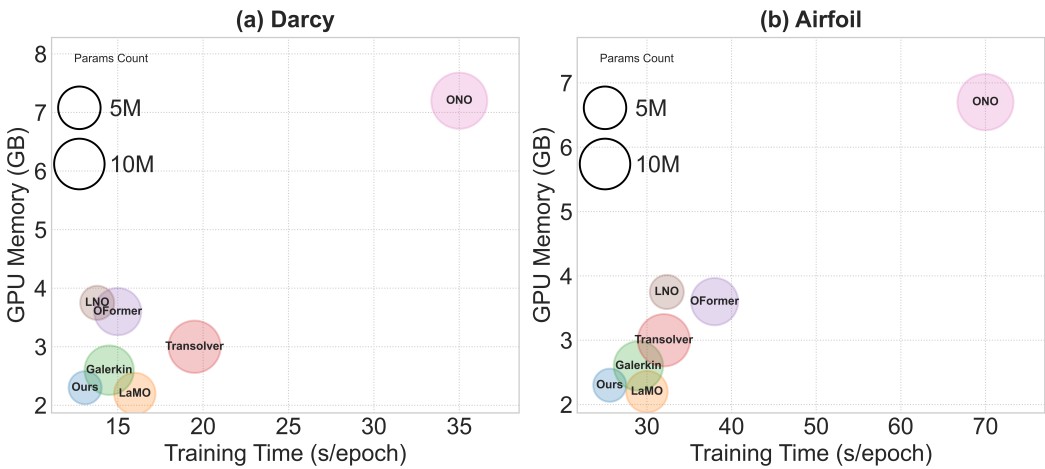

Figure 1: Comparisons of training time per epoch, number of parameters, and GPU memory among existing SOTA models on (a) Darcy and (b) airfoil, where AMO exhibits the strongest incremental gains.

## 5.3 ABLATION STUDIES

**Adaptive kernels vs. static kernels.** We now consider the need and benefits of using adaptive kernels. A kernel is adaptive when its parameterization (e.g., coefficients) varies with the input. In this work, the formulation of Equation 6 varies with the learned poles $a_{1:i}$ and thus is an adaptive kernel. We also randomly fix the value of $a_{1:i}$ for static kernels for comparison. Furthermore, although a total of $i$ poles are needed for $i$-th processing block, one can still identify more poles and select the best $i$ poles for implementation. Table 4 shows the relative $L^2$ error results across six benchmark datasets and the European options (EO) dataset. We find that, using adaptive kernels, the

Table 3: European option pricing: relative $L^2$ error and resource profile. Lower is better for error, GPU memory, and training time. Parameter counts shown in millions. **Bold** = best, underline = second best, and red = third best.

| Models | Rel. $L^2$ ($\downarrow$) | Training Time (sec/epoch, $\downarrow$) | Params (M, $\downarrow$) |
|---|---|---|---|
| FNO (Li et al., 2020) | 0.0016 | 25.1 | 3.78 |
| LNO (Wang & Wang, 2024) | 0.0010 | 21.7 | 2.56 |
| Transolver (Wu et al., 2024) | 0.0012 | 22.3 | 5.91 |
| LAMO (Tiwari et al., 2025) | 0.0008 | 22.5 | 3.52 |
| **AMO (ours)** | **0.0006** | **20.8** | **1.21** |

relative $L^2$ errors reduce significantly compared to using static poles for all benchmark problems considered. In fact, the relative $L^2$ errors when selecting only 4 poles are lower than those when selecting 32 static poles.

Table 4: Relative $L^2$ error comparisons for **Static** vs. **Adaptive** kernels across seven benchmarks. Lower is better.

| Models | Number of poles | Elasticity | Plasticity | Airfoil | Pipe | N-S | Darcy | EO |
|---|---|---|---|---|---|---|---|---|
| AMO (static) | 32 | 0.0097 | 0.0021 | 0.0067 | 0.0072 | 0.1103 | 0.0174 | 0.0035 |
| | 4 | 0.0056 | 0.0012 | 0.0033 | 0.0029 | 0.0311 | 0.0057 | 0.0014 |
| | 6 | 0.0051 | 0.0010 | 0.0031 | 0.0027 | 0.0298 | 0.0047 | 0.0010 |
| | 8 | 0.0049 | 0.0008 | 0.0027 | 0.0025 | 0.0281 | 0.0036 | 0.0009 |
| AMO (adaptive) | 16 | 0.0046 | 0.0008 | 0.0023 | 0.0028 | 0.0290 | 0.0029 | 0.0008 |
| | 32 | **0.0043** | **0.0006** | **0.0020** | **0.0023** | **0.0278** | **0.0021** | **0.0006** |
| | 64 | 0.0048 | 0.0007 | 0.0036 | 0.0031 | 0.0372 | 0.0046 | 0.0009 |

**Need for ensuring orthogonality.** To understand how orthogonal kernels affect AMO performance, we conduct another ablation study by using non-orthogonal kernels (i.e., Equation 5) in the AMO framework. In this case, the transfer functions used in SSMs are $H_i(e^{i\omega}) = \overline{(1 - |a_i|^2) \sum_{n=0}^{\infty} (\overline{a_i})^n e^{in\omega}}$ to match the output of AFD operation. Without orthogonality, AMO experiences higher relative $L^2$ error, especially for problems with irregular geometries (e.g., airfoil $0.0020 \rightarrow 0.0083$ and elasticity $0.0043 \rightarrow 0.0094$). At the same time, the training time also increases by $\sim 50.3\%$ per epoch on average across all six benchmark datasets. This shows that the use of orthogonal kernels (i.e., TM systems) helps improve both accuracy and computational efficiency of AMO solver.

**Choice of SSMs.** Finally, we evaluate the choice of bidirectional SSMs in AMO compared to unidirectional SSMs and multidirectional SSMs. Results in Figure 2 indicate that the choice of bidirectional SSMs in AMO consistently outperforms other two SSMs in all datasets.

## 5.4 EXPERIMENT USING REAL-WORLD NOISY DATASET

To validate AMO's performance on noisy real-world datasets, we perform experiments using the latex glove DIC (Digital Image Correlation) original dataset (You et al., 2022). The goal is to learn the mechanical response of a nitrile glove sample directly from experimental data, without assuming a known constitutive law. The goal is to predict the displacement field at the current loading step. The input includes the spatial coordinates, the displacement field from the previous step, and the current boundary displacement. We compare the performance of AMO to the current SOTA of this dataset, IFNO, as well as FNO as follows. To ensure fair comparison, we conduct experiments using the same settings as IFNO with the number of hidden layers ranging from 3 to 12.

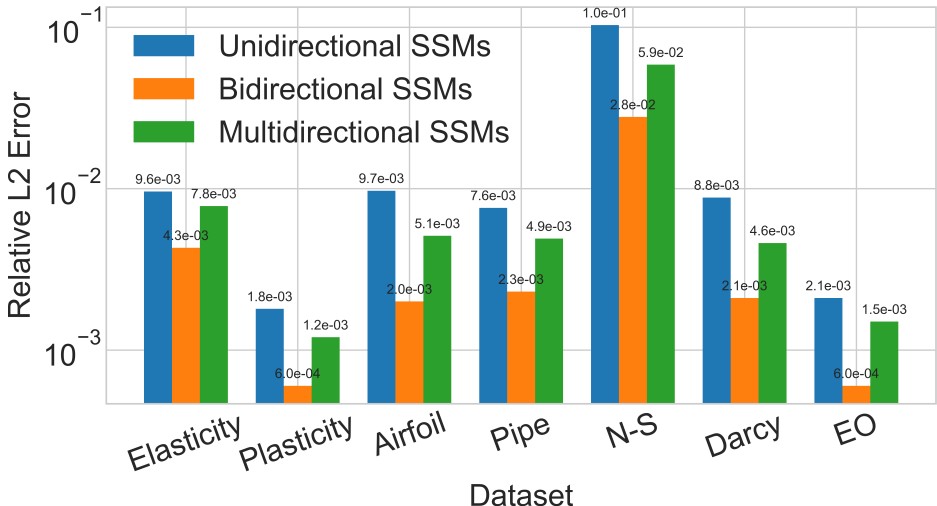

Figure 2: Contribution of three SSMs across seven benchmark datasets. Note that we do not apply weights shared for all experiments. Lower is better.

Table 5: Relative $L^2$ error of AMO and other baselines using the latex glove DIC (Digital Image Correlation) original dataset.

| Number of hidden layers | AMO | IFNO | FNO |
|---|---|---|---|
| 3 | 2.87E-02 $\pm$ 4.29E-04 | 3.43E-02 $\pm$ 4.96E-04 | 3.40E-02 $\pm$ 4.09E-04 |
| 6 | 2.50E-02 $\pm$ 3.28E-04 | 3.34E-02 $\pm$ 4.53E-04 | 3.84E-02 $\pm$ 4.21E-04 |
| 12 | 2.32E-02 $\pm$ 4.20E-04 | 3.32E-02 $\pm$ 4.41E-04 | 4.66E-02 $\pm$ 1.47E-03 |

In addition, You et al. (2022) also reported the results of generalized Mooney-Rivlin (GMR) model in two settings. The relative $L^2$ errors of GMR model fitting and GMR inverse analysis are 3.30E-01 and 2.91E-01, respectively. We can observe that our AMO consistently outperforms other models in every $L$. Finally, the best reported result of IFNO is 3.30E-02 $\pm$ 4.63E-04 when $L = 24$ (You et al., 2022). Although we do not conduct the experiment $L = 24$ due to the limited time, our AMO still performs better than the best result of IFNO.

## 6 CONCLUSIONS

In this paper, we propose a novel neural operator AMO for solving nonlinear PDEs on irregular geometries and singularities. AMO maps the physical tokens in an RKHS where the global spectral transform and data-dependent orthogonal kernels are incorporated. By conducting a tailored design of the TM layer and SSM block fully guided by the AFD theory, we show that the output of AMO exactly matches with AFD oepration, hence offering rigorous convergence guarantee and other desirable properties. We show that the novel architecture of AMO enables its outstanding performance compared to existing SOTA neural operators in a series of physical and financial benchmark problems.

## 7 REPRODUCIBILITY STATEMENT

All code and datasets have been either made publicly available in an anonymous repository or as a part of supplementary material to facilitate replication and verification. The experimental setup, including training steps, model configurations, and hardware details, is described in detail in the paper. We have also provided a full description of implementation details, to assist others in reproducing our experiments. Additionally, six benchmark datasets, such as pipe, are publicly available, ensuring consistent and reproducible evaluation results.

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

# A    NOTATION LIST

| | |
|---|---|
| $a$ | Parameter function (input) |
| $\hat{u}_{N,\theta}$ | Output of AMO with $N$ blocks and parameters $\theta$ |
| $N$ | Number of processing blocks |
| $N_s$ | Number of input physical tokens |
| $M$ | Number of encoded latent tokens ($M \ll N_s$) |
| $D_{\text{embed}}$ | Embedding dimension of latent tokens |
| $\mathbf{x}_{\text{phys}}$ | Input physical features |
| $\mathbf{g}_{\text{phys}}$ | Positional embedding of coordinates |
| $\mathbf{z}_i$ | Token representation after the $i$-th block |
| $\mathbf{z}_0$ | Encoded tokens produced by the lifting operator $\mathcal{P}$ |
| $\mathbf{z}_1$ | Tokens mapped into RKHS by operator $\mathcal{R}$ |
| $\mathcal{P}$ | Lifting operator mapping physical tokens to encoded tokens |
| $\mathcal{Q}$ | Projection operator mapping latent tokens back to output space |
| $\mathcal{R}$ | Mapping operator from latent tokens to RKHS |
| $\mathcal{L}^i$ | Processing block at layer $i$ ($\text{SSM}^i \circ \text{TM}^i$) |
| $\mathcal{S}^i$ | Aggregation operator with skip connections at block $i$ |
| $\text{TM}^i$ | TM layer performing spectral transform via TM bases |
| $\text{SSM}^i$ | Bidirectional SSM block parameterized by transfer function |
| $\mathscr{B}_i(z; a_{1:i})$ | $i$-th TM basis generated by poles $a_{1:i}$ |
| $a_{1:i}$ | Set of learned poles $\{a_1, \dots, a_i\}$ in the unit disk $\mathbb{D}$ |
| $K_a(z)$ | Reproducing kernel $\frac{1}{1-\bar{a}z}$ |
| $H_i(e^{i\omega})$ | Transfer function of the $i$-th SSM block |
| $h_i[n]$ | Impulse response of the $i$-th SSM block |
| $\langle x, f \rangle$ | Inner product $\frac{1}{\tilde{N}} \sum_{n=0}^{\tilde{N}-1} x[n]\overline{f(e^{i2\pi n/\tilde{N}})}$ |
| $\odot$ | Element-wise (Hadamard) product |
| $\mathcal{H}$ | Reproducing Kernel Hilbert Space (RKHS) |
| $\tilde{N}$ | Length of signal in inner product definition |

# B    ILLUSTRATIVE EXAMPLES

**1-D advection PDE with high-frequency perturbation.**    We evaluate LaMO on a 1-D linear advection benchmark governed by

$$u_t + c\,u_x = 0 \tag{17}$$

on a periodic unit interval. Initial conditions $u_0(x)$ are synthesized as smooth Fourier mixtures $\sum_{k=1}^{k_{\max}} a_k \sin(2\pi kx + \phi_k)$ with amplitudes decaying as $a_k \sim (1+k)^{-1}$, to which we add a weak high-frequency spike at wavenumber $k_{\text{hi}}$ to probe aliasing and phase accuracy. Trajectories are advanced to time $T$ with a conservative first-order upwind scheme at Courant number CFL $= c\,\Delta t/\Delta x \le 0.5$, ensuring stability while preserving sharp phase relationships; the target is the advected field $u(\cdot, T)$.

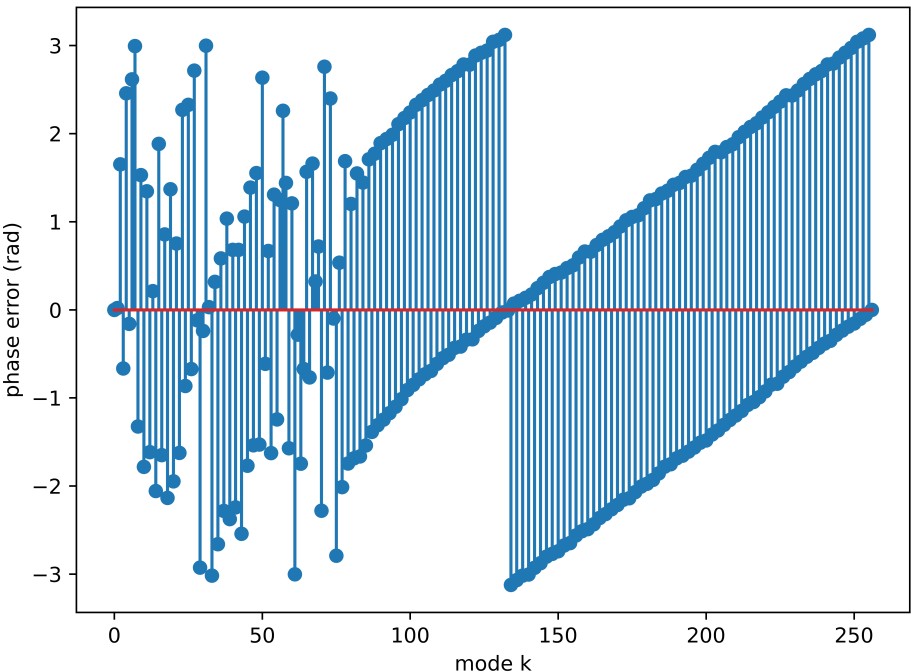

Figure 3: Phase error of solutions predicted by LaMO.

Figure 3 visualizes the phase error of LaMO's predictions, revealing a pronounced degradation for high-frequency modes (approximately $k \in [140, 250]$). This suggests that LaMO struggles to faithfully capture phase at the upper end of the spectrum.

**2-D Darcy flow equation with fractal noise.** We construct a challenging 2-D Darcy dataset by solving

$$-\nabla \cdot \big(k(x, y)\nabla u(x, y)\big) = f(x, y) \tag{18}$$

on $[0, 1]^2$ with homogeneous Dirichlet boundaries, where the permeability $k$ is positive, highly heterogeneous, and fractal-like. Specifically, $k$ is generated by exponentiating a band-limited fractional Gaussian field (small Hurst parameter for roughness) and then modulating it with narrow channel masks and inclusions to induce strong anisotropy and high contrast. The forcing $f$ combines a weak background term with several randomized Gaussian sources/sinks, which produce near-singular behavior in the solution. The variable-coefficient elliptic problem is discretized on a Cartesian grid using a flux-conservative 5-point stencil with harmonic averaging of $k$, and solved to tight tolerance via conjugate gradients. For learning, each sample is subsampled irregularly: we draw $P$ points $\{(x_i, y_i)\}$ and record $u(x_i, y_i)$, yielding pairs $(\mathrm{XY}, U)$ without exposing $k$ or $f$.

To visualize and stress singular structures, we show in Figure 4 (a) and (c): (i) contours of the potential $u$ highlighting global flow topology, and (ii) a logarithmic map of the gradient magnitude, $\log |\nabla u|$, computed on a reconstructed dense grid via triangulation. Figure 4 shows LAMO cannot capture the singularities of $u$ and $\log |\nabla u|$. Furthermore, once the complex singularities appear, the performance of LAMO will be affected.

**3-D Brusselator problem.** We introduce a new 3-D Brusselator (diffusion-reaction equation) problem using the dataset from Laplace neural operator (LNO) (Cao et al., 2024). The Brusselator problem is formulated as:

$$D\frac{\partial^2 y}{\partial x^2} + ky^2 - \frac{\partial y}{\partial t} = f(x, t), \tag{19}$$

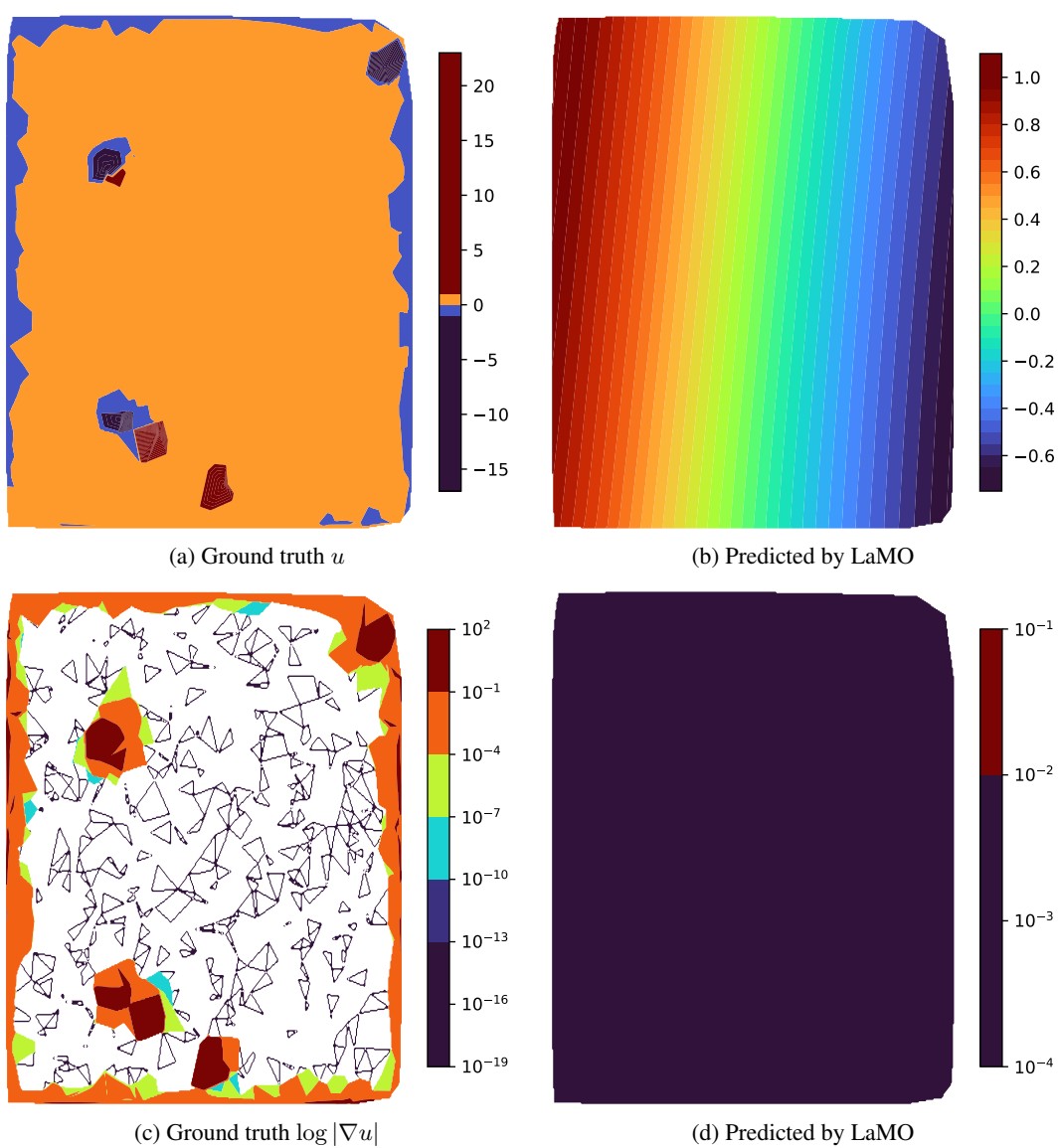

(a) Ground truth $u$

(b) Predicted by LaMO

(c) Ground truth $\log|\nabla u|$

(d) Predicted by LaMO

Figure 4: The predicted results produced by LaMO compared to the ground truth.

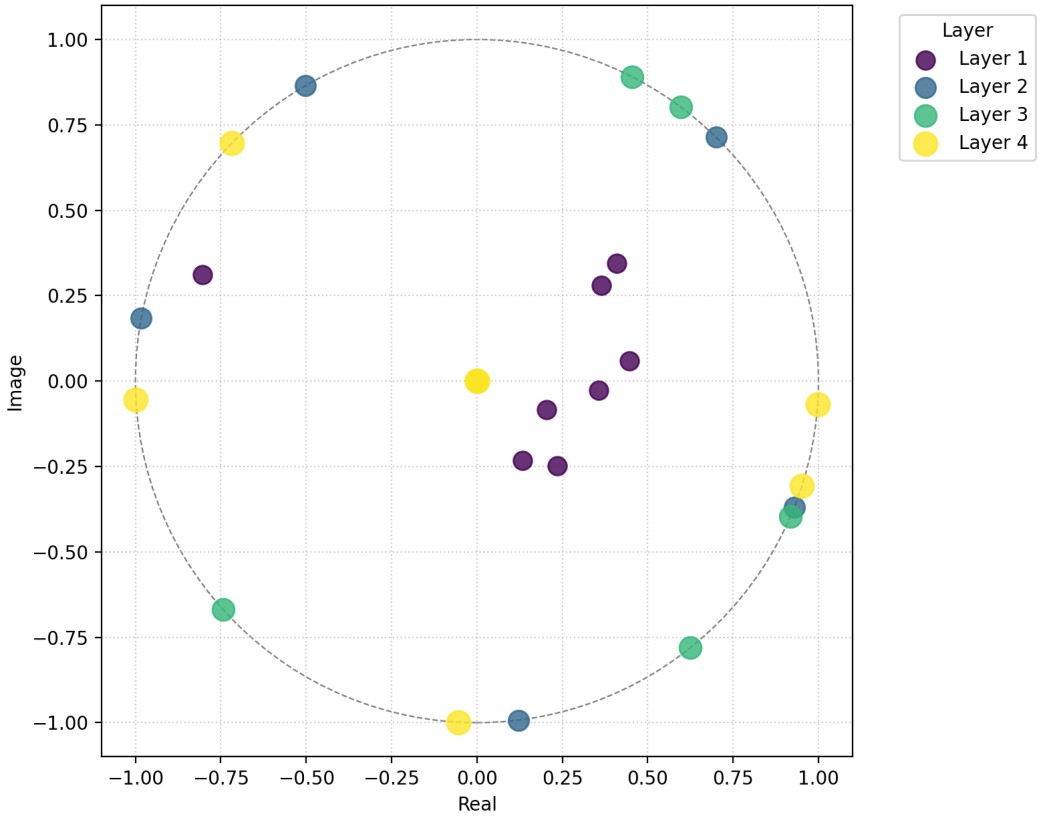

Figure 5: Learned poles distribution for the 2-D Darcy flow equation.

where $y(x,t)$ represents the concentration of chemical substances or particles at location $x$ and time $t$, $f(x,t)$ is the source term and $A$ is the amplitude of the source term. In this problem, the diffusion coefficient, $D = 0.01$, and the reaction rate, $k = 0.01$.

## C DISTRIBUTION OF SELECTED POLES REFLECTS PROBLEM CHARACTERISTICS

To understand how AMO's pole selection process is adaptive to the characteristics and nature of the problem, we illustrate the learned pole distributions for the 2-D Darcy flow problem and 3-D Brusselator problem in Figures 5. To clarify, here we give a brief overview of the visualization results: The distribution of selected poles for the 2-D Darcy flow problem is shown in Figures 5 and 6, respectively.

We observe that, across the layers, the learned poles of AMO on Darcy flow problem tend to approach to the boundary of the unit disk, while those on the Brusselator problem tend to be in the interior of the unit disk. The reason is that, Darcy flow problem is an elliptic equation, which is a smoothing operator. Thus, even though the input coefficient (the permeability) is very rough and discontinuous, the solution inside the domain will be well-behaved. Therefore, the challenging characteristics and singularities of the Darcy flow problem are located at the boundaries, and then more adaptive poles would be put there. Meanwhile, the complexity of the Brusselator problem does not come from the boundaries. It comes from the local, non-linear reaction that happens at every single point inside the domain. Therefore, most of the learned poles should be put inside the unit disk.

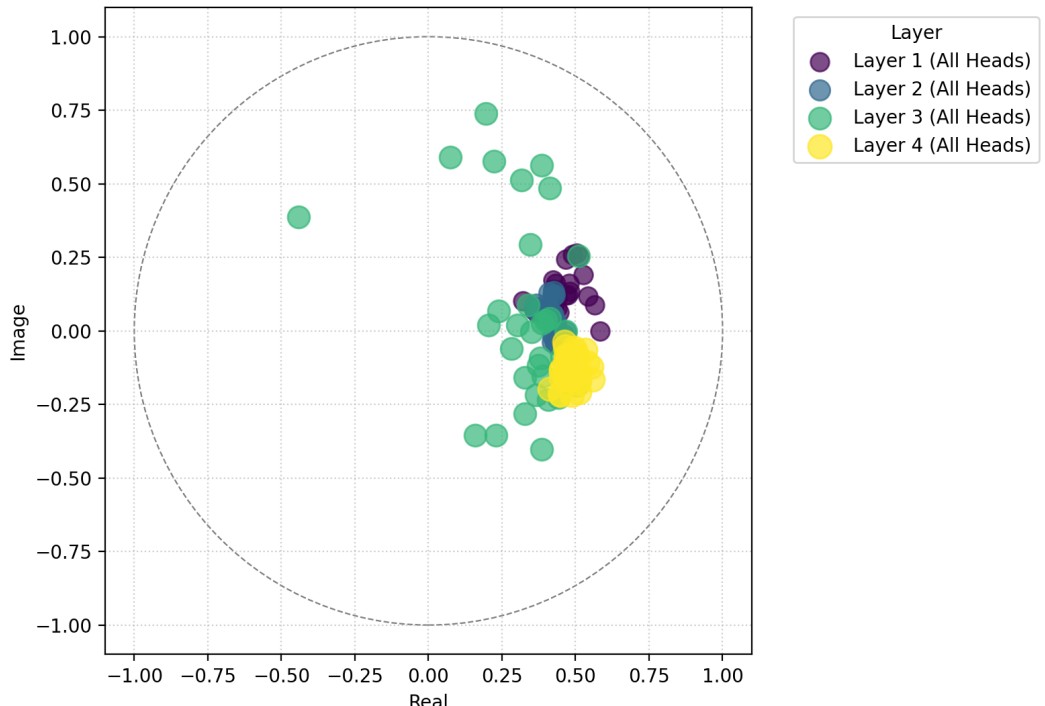

Figure 6: Learned poles distribution for the 3-D Brusselator equation.

## D  THEORETICAL RESULTS OF AMO

**Basic settings.**  Let $\mathbb{D} = \{z \in \mathbb{C} : |z| < 1\}$. Consider a reproducing kernel Hilbert space (RKHS) $(\mathcal{H}, \langle \cdot, \cdot \rangle_{\mathcal{H}})$ of complex-valued functions on $\mathbb{D}$ with the following properties.

**Assumption D.1.**  There is a family of normalized reproducing kernels $\{e_a : a \in \mathbb{D}\} \subset \mathcal{H}$ such that

$$e_a(z) = \frac{\sqrt{1 - |a|^2}}{1 - \overline{a}z} \in \mathcal{H}, \qquad \langle f, e_a \rangle_{\mathcal{H}} = f(a)\sqrt{1 - |a|^2} \quad \forall f \in \mathcal{H}, \, a \in \mathbb{D}. \tag{20}$$

Given a pole sequence $a_{1:\infty} = (a_1, a_2, \dots) \subset \mathbb{D}$, define the Takenaka–Malmquist (TM) system by

$$\mathscr{B}_1(z) = e_{a_1}(z), \qquad \mathscr{B}_i(z) = e_{a_i}(z) \prod_{j=1}^{i-1} \frac{z - a_j}{1 - \overline{a_j}z} \quad (i \geq 2). \tag{21}$$

Assume $\{\mathscr{B}_i\}_{i \geq 1}$ is an orthonormal system in $\mathcal{H}$, and its closed linear span equals the model space

$$K_B := \overline{\mathrm{span}}\{\mathscr{B}_i : i \geq 1\} \subseteq \mathcal{H}, \tag{22}$$

where $B$ is the Blaschke product with zeros $\{a_i\}$.

**AMO notation.**  Let $s \in \mathcal{H}$ be the latent target representation and $u^{\star} = \mathcal{Q}(s)$, where $\mathcal{Q} : \mathcal{H} \to \mathcal{U}$ is a Lipschitz decoder with constant $L_{\mathcal{Q}}$. Define the ideal TM coefficients and partial sums

$$c_i^{\star} := \langle s, \mathscr{B}_i \rangle_{\mathcal{H}}, \qquad s_N := \sum_{i=1}^{N} c_i^{\star} \mathscr{B}_i. \tag{23}$$

AMO learns estimates $\widehat{c}_i$ of $c_i^{\star}$ (via an SSM in the frequency domain) and aggregates them through the skip connection:

$$z_{i+1} := z_i + \widehat{c}_i \mathscr{B}_i, \qquad z_1 := 0. \tag{24}$$

## D.1 Aggregation identity and frequency-domain coefficient extraction

**Lemma D.2.** *Under 24, one has, for every $N \in \mathbb{N}$,*

$$z_{N+1} = \sum_{i=1}^{N} \widehat{c}_i \, \mathscr{B}_i. \tag{25}$$

*Proof.* The proof is by induction. For $N = 1$, $z_2 = z_1 + \widehat{c}_1 \mathscr{B}_1 = \widehat{c}_1 \mathscr{B}_1$, so 25 holds. Assume 25 holds for $N$, i.e., $z_{N+1} = \sum_{i=1}^{N} \widehat{c}_i \, \mathscr{B}_i$. Then

$$z_{N+2} = z_{N+1} + \widehat{c}_{N+1} \mathscr{B}_{N+1} = \sum_{i=1}^{N+1} \widehat{c}_i \, \mathscr{B}_i,$$

which establishes the claim for $N + 1$. $\qquad\square$

**Lemma D.3.** *Suppose the $i$-th SSM has transfer function*

$$H_i(e^{i\omega}) = \overline{\mathscr{B}_i(e^{i\omega})}, \tag{26}$$

*so that the block multiplies the input spectrum by $\overline{\mathscr{B}_i}$ and outputs the zero-lag correlation. If the discrete inner product used by AMO is a consistent quadrature for $\langle \cdot, \cdot \rangle_{\mathcal{H}}$ on the class $\{s\} \cup \{\mathscr{B}_i\}$, then*

$$\widehat{c}_i \to \langle s, \mathscr{B}_i \rangle_{\mathcal{H}} = c_i^\star \quad \text{as the quadrature is refined.} \tag{27}$$

*Proof.* By 26, the block forms (pointwise on the grid) $Y_i = \overline{\mathscr{B}_i} \cdot s$ in the transform domain; the zero-lag correlation is the discretized inner product $\langle s, \mathscr{B}_i \rangle_{\text{disc}}$. Consistency of the quadrature implies $\langle s, \mathscr{B}_i \rangle_{\text{disc}} \to \langle s, \mathscr{B}_i \rangle_{\mathcal{H}}$ as the grid is refined. Hence $\widehat{c}_i \to c_i^\star$. $\qquad\square$

## D.2 Convergence in the model space and projection error

**Theorem D.4.** *Under Assumption D.1, if AMO recovers the exact coefficients $c_i^\star = \langle s, \mathscr{B}_i \rangle_{\mathcal{H}}$, then*

$$s_N := \sum_{i=1}^{N} c_i^\star \mathscr{B}_i \xrightarrow[N \to \infty]{\mathcal{H}} \Pi_{K_B} s, \tag{28}$$

*the orthogonal projection of $s$ onto $K_B$. Consequently,*

$$\|u^\star - \mathcal{Q}(s_N)\| \le L_{\mathcal{Q}} \|s - \Pi_{K_B} s\|_{\mathcal{H}} + L_{\mathcal{Q}} \|\Pi_{K_B} s - s_N\|_{\mathcal{H}} \xrightarrow[N \to \infty]{} L_{\mathcal{Q}} \operatorname{dist}(s, K_B). \tag{29}$$

*Proof.* Because $\{\mathscr{B}_i\}$ is an orthonormal basis (ONB) of $K_B$, the Fourier expansion of $\Pi_{K_B} s$ in this ONB has coefficients $\langle s, \mathscr{B}_i \rangle_{\mathcal{H}}$, and the $N$-th partial sum equals $s_N$. Convergence in norm to the projection is standard for orthogonal series in a Hilbert space, giving 28. The bound 29 follows from Lipschitz continuity of $\mathcal{Q}$:

$$\|u^\star - \mathcal{Q}(s_N)\| = \|\mathcal{Q}(s) - \mathcal{Q}(s_N)\| \le L_{\mathcal{Q}} \|s - s_N\| \le L_{\mathcal{Q}} \big( \|s - \Pi_{K_B} s\| + \|\Pi_{K_B} s - s_N\| \big).$$

$\qquad\square$

*Remark.* No greedy or maximal selection is used. The MLP-generated poles determine $K_B$; AMO converges to $\Pi_{K_B} s$, and to $s$ whenever $s \in K_B$.

## D.3 Best $N$-term error and rates without greedy selection

**Definition D.5.** *Let $\mathcal{D} := \{\mathscr{B}_i(\cdot; a_{1:i}) : a_{1:i} \in \mathbb{D}^i, \, i \in \mathbb{N}\}$ be the TM dictionary. Define the best $N$-term error*

$$E_N(s) := \inf_{a_{1:N}, c_{1:N}} \Big\| s - \sum_{i=1}^{N} c_i \, \mathscr{B}_i(\cdot; a_{1:i}) \Big\|_{\mathcal{H}}. \tag{30}$$

**Theorem D.6.** *Let $\tilde{a}_{1:N}$ be the poles output by the MLP and set $c_i^{\star} = \langle s, \mathscr{B}_i(\cdot; \tilde{a}_{1:i}) \rangle_{\mathcal{H}}$. If AMO learns $\widehat{c}_i$, then*

$$\left\| s - \sum_{i=1}^{N} \widehat{c}_i \, \mathscr{B}_i(\cdot; \tilde{a}_{1:i}) \right\|_{\mathcal{H}} \le E_N(s) + \Delta_{pole}(N) + \left( \sum_{i=1}^{N} |\widehat{c}_i - c_i^{\star}|^2 \right)^{\frac{1}{2}}, \tag{31}$$

*where*

$$\Delta_{pole}(N) := \inf_{c_{1:N}} \left\| s - \sum_{i=1}^{N} c_i \, \mathscr{B}_i(\cdot; \tilde{a}_{1:i}) \right\|_{\mathcal{H}} - E_N(s) \ge 0. \tag{32}$$

*Proof.* Choose $a_{1:N}^{\text{best}}, c_{1:N}^{\text{best}}$ that attain (or $\varepsilon$-attain) $E_N(s)$ and denote $s_N^{\text{best}} := \sum_{i=1}^{N} c_i^{\text{best}} \mathscr{B}_i(\cdot; a_{1:i}^{\text{best}})$. Then

$$\left\| s - \sum_{i=1}^{N} \widehat{c}_i \mathscr{B}_i(\cdot; \tilde{a}_{1:i}) \right\| \le \left\| s - s_N^{\text{best}} \right\| + \left\| s_N^{\text{best}} - \sum_{i=1}^{N} c_i^{\star} \mathscr{B}_i(\cdot; \tilde{a}_{1:i}) \right\| + \left\| \sum_{i=1}^{N} (c_i^{\star} - \widehat{c}_i) \mathscr{B}_i(\cdot; \tilde{a}_{1:i}) \right\|$$

$$\le E_N(s) + \Delta_{\text{pole}}(N) + \left( \sum_{i=1}^{N} |c_i^{\star} - \widehat{c}_i|^2 \right)^{1/2}.$$

The last inequality uses the definition of $\Delta_{\text{pole}}(N)$ and orthonormality of $\{\mathscr{B}_i(\cdot; \tilde{a}_{1:i})\}_{i=1}^{N}$. □

**Corollary D.7.** *Assume for the fixed MLP-produced poles $\tilde{a}_{1:i}$ that the exact TM coefficients satisfy the weak-$\ell^p$ decay*

$$|c_i^{\star}|^* \le C \, i^{-1/p}, \qquad 0 < p < 2,$$

*where $(|c_i^{\star}|^*)$ is the nonincreasing rearrangement. Then*

$$\inf_{c_{1:N}} \left\| s - \sum_{i=1}^{N} c_i \, \mathscr{B}_i(\cdot; \tilde{a}_{1:i}) \right\|_{\mathcal{H}} = \mathcal{O}\big(N^{\frac{1}{2} - \frac{1}{p}}\big). \tag{33}$$

*If, in addition, $\Delta_{pole}(N) = o(1)$ and $\left( \sum_{i=1}^{N} |\widehat{c}_i - c_i^{\star}|^2 \right)^{1/2} = o(1)$, then the AMO error in 31 is $\mathcal{O}\big(N^{\frac{1}{2} - \frac{1}{p}}\big).$*

*Proof.* For an orthonormal system, the best $N$-term error equals the $\ell^2$ tail of the rearranged coefficients. With $|c_i^{\star}|^* \le C i^{-1/p}$ and $p < 2$,

$$\sum_{i>N} (|c_i^{\star}|^*)^2 \le C^2 \sum_{i>N} i^{-2/p} = \mathcal{O}\big(N^{1 - \frac{2}{p}}\big),$$

hence the norm error (square root) is $\mathcal{O}(N^{\frac{1}{2} - \frac{1}{p}})$. □

## D.4 LEARNING AND DISCRETIZATION ERRORS

**Assumption D.8.** *Each $\widehat{c}_i$ is obtained by ERM over $m$ i.i.d. frequency samples using a hypothesis class with effective capacity $d_{\text{eff}}$ under sub-Gaussian noise, so that*

$$\mathbb{E}\big[|\widehat{c}_i - c_i^{\star}|\big] = \mathcal{O}\Big(\sqrt{\tfrac{d_{\text{eff}}}{m}}\Big). \tag{34}$$

**Lemma D.9.** *Let $\langle \cdot, \cdot \rangle_{\tilde{N}}$ be a discrete inner product (e.g., uniform frequency grid) that is a consistent quadrature for $\langle \cdot, \cdot \rangle_{\mathcal{H}}$ on the class generated by $\{s\} \cup \{\mathscr{B}_i\}$. Then there exists $\varepsilon_{\text{disc}}(\tilde{N}) \downarrow 0$ such that*

$$\big| \langle f, g \rangle_{\mathcal{H}} - \langle f, g \rangle_{\tilde{N}} \big| \le \varepsilon_{\text{disc}}(\tilde{N}) \qquad \text{for all } f \in \{s\}, \, g \in \{\mathscr{B}_i\}_{i \ge 1}. \tag{35}$$

*Proof.* Since point evaluations are continuous linear functionals in an RKHS and the involved functions are continuous on compact subsets, standard quadrature consistency yields 35. (If $f, g$ are analytic in an annulus around the unit circle, one gets exponential rates; under Sobolev regularity, algebraic rates.) □

**Theorem D.10.** *Under Assumptions D.1 and D.8 and Lemma D.9, the AMO output after $N$ blocks and $\tilde{N}$ grid points satisfies*

$$\|u^\star - \hat{u}_{N,\theta}\| \leq L_{\mathcal{Q}}\Big(E_N(s) + \Delta_{pole}(N) + \Big(\sum_{i=1}^{N}|\hat{c}_i - c_i^\star|^2\Big)^{1/2}\Big) + \varepsilon_{\mathrm{disc}}(\tilde{N}), \quad (36)$$

*with $\mathbb{E}[|\hat{c}_i - c_i^\star|] = \mathcal{O}(\sqrt{d_{\mathrm{eff}}/m})$ and $\varepsilon_{\mathrm{disc}}(\tilde{N}) \to 0$ as $\tilde{N} \to \infty$.*

*Proof.* Apply Theorem D.6 to bound the latent $\mathcal{H}$-error. Then use Lipschitz continuity of $\mathcal{Q}$ to transfer the bound to the output space. The discretization error adds $\varepsilon_{\mathrm{disc}}(\tilde{N})$ due to 35. $\qquad \square$

### D.5 STABILITY TO POLE PERTURBATIONS

**Lemma D.11.** *For $a, b \in \mathbb{D}$ and $z \in \mathbb{D}$,*

$$\left|\frac{1}{1 - \bar{a}z} - \frac{1}{1 - \bar{b}z}\right| \leq \frac{|a - b|}{(1 - |a|)(1 - |b|)}, \quad (37)$$

$$\left|\sqrt{1 - |a|^2} - \sqrt{1 - |b|^2}\right| \leq \frac{|a - b|}{\sqrt{1 - \max\{|a|, |b|\}^2}}, \quad (38)$$

*and for $F(z; a) = \dfrac{z - a}{1 - \bar{a}z}$,*

$$|F(z; a) - F(z; b)| \leq \frac{4|a - b|}{(1 - |a|)(1 - |b|)}, \qquad |F(z; a)| \leq 1. \quad (39)$$

*Proof.* For 37,

$$\frac{1}{1 - \bar{a}z} - \frac{1}{1 - \bar{b}z} = \frac{(\bar{a} - \bar{b})z}{(1 - \bar{a}z)(1 - \bar{b}z)},$$

and $|1 - \bar{a}z| \geq 1 - |a||z| \geq 1 - |a|$, $|z| \leq 1$, yielding the bound. For 38, use the mean-value theorem on $x \mapsto \sqrt{1 - x}$ with $x = |a|^2, |b|^2$ and $||a|^2 - |b|^2| \leq |a - b|(|a| + |b|) \leq 2|a - b|$. For 39, expand

$$F(z; a) - F(z; b) = \frac{(b - a) + (\bar{a} - \bar{b})z^2 + (a\bar{b} - b\bar{a})z}{(1 - \bar{a}z)(1 - \bar{b}z)},$$

and bound the numerator by $C|a - b|$ for $|z| \leq 1$, while the denominator is bounded below by $(1 - |a|)(1 - |b|)$. $\qquad \square$

**Theorem D.12.** *Let $a_{1:i}, \tilde{a}_{1:i} \in \mathbb{D}$ with $|\tilde{a}_j - a_j| \leq \delta_j$. Then there exist constants $C_i > 0$ (depending on $a_{1:i}$) such that*

$$\|\mathscr{B}_i(\cdot; \tilde{a}_{1:i}) - \mathscr{B}_i(\cdot; a_{1:i})\|_{\mathcal{H}} \leq C_i \sum_{j=1}^{i} \frac{\delta_j}{1 - |a_j|}. \quad (40)$$

*Consequently, for any coefficients $\hat{c}_i$,*

$$\Big\|\sum_{i=1}^{N} \hat{c}_i \, \mathscr{B}_i(\cdot; \tilde{a}_{1:i}) - \sum_{i=1}^{N} \hat{c}_i \, \mathscr{B}_i(\cdot; a_{1:i})\Big\|_{\mathcal{H}} \leq \Big(\sum_{i=1}^{N} |\hat{c}_i| \, C_i\Big)\Big(\sum_{j=1}^{N} \frac{\delta_j}{1 - |a_j|}\Big). \quad (41)$$

*Proof.* Write

$$\mathscr{B}_i(\cdot; a_{1:i}) = e_{a_i} \prod_{j=1}^{i-1} F(\cdot; a_j), \qquad \mathscr{B}_i(\cdot; \tilde{a}_{1:i}) = e_{\tilde{a}_i} \prod_{j=1}^{i-1} F(\cdot; \tilde{a}_j).$$

Use the product telescoping identity

$$\prod_{k=1}^{i} P_k - \prod_{k=1}^{i} Q_k = \sum_{k=1}^{i} \Big(\prod_{j<k} P_j\Big)(P_k - Q_k)\Big(\prod_{j>k} Q_j\Big),$$

with $P_1 = e_{\tilde{a}_i}$, $Q_1 = e_{a_i}$, and $P_k = F(\cdot; \tilde{a}_{k-1})$, $Q_k = F(\cdot; a_{k-1})$ for $k \geq 2$. Taking sup-norms on $\mathbb{D}$ and using $|F(\cdot; a)| \leq 1$,

$$\|\mathscr{B}_i(\cdot; \tilde{a}_{1:i}) - \mathscr{B}_i(\cdot; a_{1:i})\|_\infty \leq \|e_{\tilde{a}_i} - e_{a_i}\|_\infty + \sum_{j=1}^{i-1} \|F(\cdot; \tilde{a}_j) - F(\cdot; a_j)\|_\infty.$$

Apply Lemma D.11 to bound each term by a constant times $\delta_j / (1 - |a_j|)$. Since evaluation functionals are continuous and the kernel is bounded on compact subsets, there exists an embedding constant $C_{\mathrm{emb}}$ with $\|f\|_{\mathcal{H}} \leq C_{\mathrm{emb}} \|f\|_\infty$ on the set considered; thus 40 follows with $C_i$ absorbing all constants. Finally,

$$\left\| \sum_{i=1}^N \widehat{c}_i \big( \mathscr{B}_i(\cdot; \tilde{a}_{1:i}) - \mathscr{B}_i(\cdot; a_{1:i}) \big) \right\|_{\mathcal{H}} \leq \sum_{i=1}^N |\widehat{c}_i| \, \|\mathscr{B}_i(\cdot; \tilde{a}_{1:i}) - \mathscr{B}_i(\cdot; a_{1:i})\|_{\mathcal{H}},$$

giving 41. □

## D.6 END-TO-END CONVERGENCE WITHOUT GREEDY SELECTION

**Theorem D.13.** *Assume:*

1. $s \in K_B$;

2. $\sum_{i=1}^\infty \mathbb{E}[|\widehat{c}_i - c_i^\star|^2]^{1/2} < \infty$ *(as sample size $m \to \infty$ and model capacity increase)*;

3. $\varepsilon_{\mathrm{disc}}(\tilde{N}) \to 0$ *as $\tilde{N} \to \infty$.*

*Then*

$$\lim_{N \to \infty} \|u^\star - \hat{u}_{N,\theta}\| = 0.$$

*Proof.* Since $s \in K_B$ and $\{\mathscr{B}_i\}$ is an ONB of $K_B$, Theorem D.4 gives $s_N \to s$ in $\mathcal{H}$. In 36, for this fixed pole sequence one has $E_N(s) = \Delta_{\mathrm{pole}}(N) = 0$. Using (2) and (3), we obtain $\|u^\star - \hat{u}_{N,\theta}\| \to 0$. □

## D.7 CONNECTION OF SSM TO CORRELATION AND AMO OUTPUT

**Proposition D.14.** *With $H_i(e^{i\omega}) = \overline{\mathscr{B}_i(e^{i\omega})}$, the $i$-th SSM block computes $\widehat{c}_i \approx \langle z_i, \mathscr{B}_i \rangle_{\mathcal{H}}$. Hence, by Lemma D.2, after $N$ blocks*

$$z_{N+1} = \sum_{i=1}^N \widehat{c}_i \, \mathscr{B}_i, \qquad \hat{u}_{N,\theta} = \mathcal{Q}(z_{N+1}). \tag{42}$$

*Proof.* The coefficient claim follows from Lemma D.3 applied to $z_i$ in place of $s$. The aggregation identity is Lemma D.2. The last equality is the definition of $\mathcal{Q}$. □

**Corollary D.15.** *All latent-space error bounds transfer to the PDE output space via*

$$\|u^\star - \hat{u}_{N,\theta}\| \leq L_{\mathcal{Q}} \left\| s - \sum_{i=1}^N \widehat{c}_i \mathscr{B}_i \right\| + \varepsilon_{\mathrm{disc}}(\tilde{N}).$$

## E TRANSFER FUNCTION

We consider a (finite) Blaschke product

$$H(z) = \prod_{j=1}^n \frac{1 - p_j z}{z - p_j}, \qquad |p_j| < 1, \tag{43}$$

and convert it into a single ratio of polynomials whose coefficients match the parameterization used to train SSMs.

**Polynomial expansion and $z^{-1}$ form.** Denote numerator and denominator polynomials

$$B_{\text{poly}}(z) = \prod_{j=1}^{n}(z - p_j), \qquad A_{\text{poly}}(z) = \prod_{j=1}^{n}(1 - p_j z), \tag{44}$$

so that $H(z) = \frac{A_{\text{poly}}(z)}{B_{\text{poly}}(z)}$. Let $d = \deg B_{\text{poly}} = \deg A_{\text{poly}} = n$. To obtain the form with a unit constant term in the denominator, divide numerator and denominator by $z^d$ and then normalize:

$$\widetilde{H}(z) \;=\; \frac{\sum_{k=0}^{d} \alpha_k z^{-k}}{\sum_{k=0}^{d} \beta_k z^{-k}} \quad\xrightarrow{\text{normalize}}\quad h_0 \;+\; \sum_{k=1}^{d} \frac{b_k}{1}\, z^{-k} \;\Big/\; \Big(1 \;+\; \sum_{k=1}^{d} a_k z^{-k}\Big). \tag{45}$$

The SSM coefficients are then reduced as:

$$h_0 = \frac{\alpha_0}{\beta_0}, \qquad b_k = \frac{\alpha_k}{\beta_0}, \qquad a_k = \frac{\beta_k}{\beta_0}, \quad k = 1, \dots, d.$$

**Example ($n = 2$).** With $p_1, p_2 \in \mathbb{C}$, expand

$$B_{\text{poly}}(z) = (z - p_1)(z - p_2) = z^2 - (p_1 + p_2)z + p_1 p_2,$$

$$A_{\text{poly}}(z) = (1 - p_1 z)(1 - p_2 z) = 1 - (p_1 + p_2)z + (p_1 p_2)z^2.$$

Divide by $z^2$ to get polynomials in $z^{-1}$ and normalize by the denominator's constant term ($\beta_0 = p_1 p_2$), yielding

$$H(z) \;=\; \frac{1 - (p_1 + p_2)z^{-1} + (p_1 p_2)z^{-2}}{p_1 p_2 - (p_1 + p_2)z^{-1} + z^{-2}} \;=\; \frac{h_0 + b_1 z^{-1} + b_2 z^{-2}}{1 + a_1 z^{-1} + a_2 z^{-2}},$$

with

$$h_0 = \frac{1}{p_1 p_2}, \quad b_1 = -\frac{p_1 + p_2}{p_1 p_2}, \quad b_2 = 1, \qquad a_1 = -\frac{p_1 + p_2}{p_1 p_2}, \quad a_2 = \frac{1}{p_1 p_2}.$$

**Efficient computation for large $n$.** Direct symbolic expansion scales poorly. Instead, we multiply degree-1 polynomials using FFT-based convolution. Represent each factor by its coefficient vector:

$$(z - p_j) \;\leftrightarrow\; [1, \, -p_j], \qquad (1 - p_j z) \;\leftrightarrow\; [1, \, -p_j],$$

and iteratively convolve to form $B_{\text{poly}}$ and $A_{\text{poly}}$. By the convolution theorem, polynomial multiplication is element-wise in the frequency domain, giving $\mathcal{O}(d \log d)$ complexity. After both polynomials are assembled, convert to $z^{-1}$ by dividing by $z^d$, then normalize by the denominator's constant term to obtain $(h_0, \{a_k\}, \{b_k\})$ as in 45.

