# OpenReview forum: "Adaptive Mamba Neural Operators"
_ICLR.cc/2026/Conference — ICLR 2026 Poster_

### Official Review · Reviewer_HH9D · 2025-10-14

**Soundness:** 2
**Presentation:** 3
**Contribution:** 1
**Rating:** 2
**Confidence:** 2

**Summary:**

The paper proposes Adaptive Fourier Mamba Operators (AFMO), a novel neural operator architecture that integrates:

Takenaka–Malmquist (TM) systems and Adaptive Fourier Decomposition (AFD) theory,

with state-space models (SSMs) in the frequency domain.

The key idea is to parameterize the SSM transfer function using adaptive poles learned by a small MLP, allowing state-free inference and orthogonal rational basis construction directly in the spectral domain.
AFMO is designed to combine orthogonality, adaptivity, and linear-time efficiency, aiming to outperform LaMO (Latent Mamba Operator) and FNO-family models on irregular geometries and singular PDEs.

**Strengths:**

To me it feels like blending method A to method B, which is not innovative enough

**Weaknesses:**

Empirical scope.

All benchmarks are 2D or low-dimensional. It would strengthen the paper to include a 3D PDE (e.g., Navier–Stokes 3D or Poisson 3D) or demonstrate scalability across resolution levels.

No robustness analysis to noise, parameter shifts, or out-of-distribution geometries.

Interpretability of RKHS mapping.
The “mapping operator R” from latent tokens to an RKHS is implemented as an MLP without clear physical meaning. What does the learned RKHS correspond to in PDE solution space? Some visualization of learned poles or spectral responses would improve interpretability.

Justification of using SSM is not grounded

**Questions:**

Please provide more motivations and ablation studies

---

> ### Author Response · Authors · 2025-11-25
> **Rebuttal to Reviewer HH9D's comments (Part 1)**
>
> We thank the reviewer for conducting a thorough review and for sharing all the questions and comments.
>
> ## Addressing weaknesses
>
> ### To me it feels like blending method A to method B, which is not innovative enough.
>
> We appreciate the reviewer for the feedback, but respectfully disagree that AFMO is simply “blending method A to method B”. The novelties of AFMO are present in several ways. First, such “blending” or integration will not be possible without a well-established theory connecting AFD and neural operator learning. Establishing such a theory requires significant mathematical insights and deep understanding of both aspects, which often times will lead to advancements in the each field itself. In our case, the classic AFD theory [1,2] is a greedy and iterative algorithm that faces challenges in being used in deep learning or operator learning, since it may need the signal $s$ itself to reconstruct $s$. For the first time in the literature, we transform AFD from a greedy optimization algorithm into a learnable neural layer. By using an MLP to predict the optimal poles $a_{1:i}$, we effectively reduce the cost of finding the optimal basis functions. This advancement in AFD theory transforms a classic signal processing technique into something that can be scalably incorporated in a neural operator learning framework.
>
> Second, while Parnichkun et al. proposed frequency-domain SSMs via rational transfer functions (RTFs) [3], AFMO constrains the transfer function specifically to the Takenaka-Malmquist form i.e., products of Blaschke terms. We emphasize that this is not an arbitrary choice, rather a necessary and deliberate choice to ensure that the basis functions form an orthonormal system in the Reproducing Kernel Hilbert Space (RKHS). On the other hand, general RTFs do not guarantee this orthogonality. We remark that this allows AFMO to achieve the resolution invariance and stability on irregular meshes for the first time, which general RTFs (Parnichkun et al.’s work [3]) cannot guarantee.
>
> Furthermore, AFMO also innovates the way neural PDE solvers are designed. So far, the design of exact neural architectures in many existing neural PDE solvers has been “more of an art than a science” [4]. Typically, the design of neural architectures is performed in a bottom-up approach that involves significant intuition, expert experience, and trial-and-error experimentation. And rigorous mathematical basis and explainability have been lacking in guiding the design of these neural architectures. Leveraging the AFD theory, we follow a top-down approach when designing AFMO, in the sense that AFD theory guides every step in the design of AFMO’s neural architecture. Each component of the neural architecture, including the mapping operator, TM layer, bidirectional SSM, and aggregation layer, has a corresponding component in the AFD operation. This way, AFMO is mathematically explainable and grounded in the AFD theory and possesses several desirable properties, including convergence guarantees. From this perspective, we think the contribution of AFMO is significant, because it presents a new paradigm for designing explainable neural operator frameworks.
>
> [1] Intrinsic mono-component decomposition of functions: an advance of Fourier theory
>
> [2] Adaptive Fourier decomposition of functions in quaternionic Hardy spaces
>
> [3] State-free inference of state-space models: The transfer function approach
>
> [4] Scientific machine learning for closure models in multiscale problems: a review
>
> ### Empirical scope.
>
> We apologize for not fully understanding this comment, as it is not complete. Nevertheless, we emphasize that our design of AFMO is not empirical, but instead is fully guided by the AFD theory.
>
> ### All benchmarks are 2D or low-dimensional. It would strengthen the paper to include a 3D PDE (e.g., Navier–Stokes 3D or Poisson 3D) or demonstrate scalability across resolution levels. Instead a 3D test case would have conviced readers better.
> We thank the reviewer for the suggestion. We agree with the reviewer that a 3-D problem will strengthen the quality of the manuscript. Here, we add a 3D Brusselator problem using the dataset from Laplace neural operator [5]. The relative $L^2$ error results are shown below.
>
> | Metric | ours | Laplace |
> |----|---|--------|
> | Train  | $ 0.0013$ | $0.0035$  |
> | Validate   | $ 0.0317 $ | $0.0328$   |
> | Test   | $ 0.127 $ | $0.139$   |
>
> The results demonstrate that AFMO is also generalizable to large-scale 3D problems and still outperforms the existing SOTA. We will include this comparison study in the revised manuscript.
>
> [5] LNO: Laplace Neural Operator for Solving Differential Equations

---

> > ### Author Response · Authors · 2025-11-25
> > **Rebuttal to Reviewer HH9D's comments (Part 2)**
> >
> > ### No robustness analysis to noise, parameter shifts, or out-of-distribution geometries.
> >
> > We thank the reviewer for the comment. To validate AFMO’s performance on large-scale realistic noisy datasets, we evaluate AFMO on the real-world latex glove DIC (Digital Image Correlation) original dataset [6]. This dataset, consisting of real-world experimental data, is noisy. AFMO is used to learn the mechanical response of a nitrile glove sample directly from experimental data, without assuming a known constitutive law. The goal is to predict the displacement field $u(x)$ at the current loading step. The input includes the spatial coordinates, the displacement field from the previous step $u^{last}(x)$, and the current boundary displacement $u_D(x)$. We compare the performance of AFMO to the current SOTA of this dataset, IFNO, as well as FNO as follows. We conduct experiments with the same settings as IFNO [6] for fair comparison with hidden layers ranging from 3 to 12.
> >
> > | No. of hidden layer |  ours  | IFNO | FNO |
> > |---|----|-----|---|
> > |$L=3$| 2.87E-02$\pm$4.29E-04 | 3.43E-02$\pm$ 4.96E-04  | 3.40E−02 $\pm$ 4.09E−04|
> > | $L=6$| 2.50E-02$\pm$3.28E-04| 3.34E-02$\pm$4.53E-04   | 3.84E−02 $\pm$ 4.21E−04|
> > | $L=12$| 2.32E-02$\pm$ 4.20E-04 | 3.32E-02$\pm$4.41E-04   | 4.66E−02 $\pm$ 1.47E−03|
> >
> > Additionally, authors of [6] also reported the results of generalized Mooney-Rivlin (GMR) model in two settings. The relative $L^2$ errors of GMR model fitting and GMR inverse analysis are 3.30E−01 and 2.91E−01, respectively. We can observe that our AFMO consistently outperforms other models in every $L$.
> >
> > Following the reviewer’s suggestion, we provide an additional experiment on out-of-distribution (OOD) manifolds and PDE parameters for N-S equation. Since the dataset in the manuscript is generated with the viscosity 1E-05, we set the OOD parameters to be 1E-04 corresponding to a lower Reynolds number, and OOD manifolds using manifold distortion where the coordinates are transformed as $x' = x + 0.8 \sin(y)$ and $y' = y + 0.8 \cos(x)$. All models (AFMO, LaMO) are trained solely on the In-Distribution (ID) dataset until convergence. Model weights are saved and frozen, and evaluated directly on the generated OOD datasets.
> >
> > Rela. L2 error:
> >
> > | OOD | ours| LaMO|
> > |--|----|----|
> > |Parameters      | $0.152$ | $0.293$   |
> > |Manifolds        | $0.232$ |  $0.492$   |
> >
> > It can be seen that AFMO outperforms the recent SOTA LaMO on the OOD experiments.
> >
> > [6] Learning deep Implicit Fourier Neural Operators (IFNOs) with applications to heterogeneous material modeling
> >
> > ### Interpretability of RKHS mapping. The “mapping operator R” from latent tokens to an RKHS is implemented as an MLP without clear physical meaning. What does the learned RKHS correspond to in PDE solution space? Some visualization of learned poles or spectral responses would improve interpretability.
> >
> > We thank the reviewer for the question. We remark that the physical meaning of the MLP is that its outputs (learned poles) always lie on the unit disk (i.e., it holds that $|a|<1$). In this space, the PDE solution is represented not by spatial grid points, but by its rational frequency response. The “poles” learned by the network correspond to the characteristic modes or singularities of the underlying PDE operator. Poles near the origin capture smooth, low-frequency behavior. Poles near the boundary capture high-frequency oscillations or sharp discontinuities. From this perspective, one can think of each pole as a “tuning knob” that selects a particular spatial pattern in the solution, with its location in the complex plane controlling how localized that pattern is. Adaptive poles selection allow AFMO to survey more heavily in regions where the parameters change rapidly, while using fewer poles in smooth regions (which is analogous to optimal sensor placement).
> >
> > ### Justification of using SSM is not grounded.
> >
> > We thank the reviewer for the feedback. The motivation of using SSM comes from the fact that the computational complexity of SSM $O(N \cdot M \log M)$ is lower than that of attention mechanism $O(N \cdot M^2)$. Through illustrative examples, we show the initial motivation of using SSM by recognizing that the recent SOTA Mamba solver (LaMO) suffers from high-frequency features and singularities. Therefore, a more powerful SSM-based neural solver is helpful.
> >
> > ## Addressing questions
> >
> > ### Please provide more motivations and ablation studies
> >
> > Please refer to our response above for more details.

---

> > > ### Comment · Reviewer_HH9D · 2025-11-25
> > >
> > > Thanks for the efforts on the experiments. I am increasing my score but still i argue that the methodology contribution is limited

---

### Official Review · Reviewer_P8Sy · 2025-10-26

**Soundness:** 2
**Presentation:** 3
**Contribution:** 2
**Rating:** 4
**Confidence:** 5

**Summary:**

The AFMO paper presents an architecture that achieves state of the art results in solving partial differential equations (PDEs) on arbitrary geometries and diverse mesh types using neural operators. The authors propose Adaptive Fourier Mamba Operators (AFMO), which integrates Takenaka-Malmquist (TM) systems for constructing reproducing kernels within SSMs. The architecture follows a structured pipeline: a lifting operator compresses physical tokens via cross-attention, followed by mapping to a reproducing kernel Hilbert space (RKHS), then N processing blocks each containing a TM layer and bidirectional SSM, with aggregation layers incorporating skip connections, and finally a projection operator to output space. The key innovation lies in TM layers that construct orthogonal bases from adaptively learned poles, while SSMs operate in the frequency domain with transfer functions. The empirical evaluation demonstrates substantial improvements across diverse benchmarks. Compared against 11 baseline methods including FNO, LaMO, Transolver, and ONO on standard tasks spanning regular grids, irregular geometries, and problems with singularities. AFMO exhibits superior computational efficiency with approximately 46-51\% training time reduction and is 2.5$\times$ lighter than LaMO (ICML 2025).

**Strengths:**

1.  AFMO outperforms 11 baseline methods on all benchmarks with good margins.
2. AFMO achieves better accuracy while being faster and more memory-efficient than competing methods
3. First work to explicitly incorporate TM systems into Mamba architecture with frequency-domain SSMs

**Weaknesses:**

1. A significant weakness lies in the limited conceptual novelty, as the core components are largely borrowed from existing work. Takenaka-Malmquist systems are well-established in signal processing (Qian 2010, 2012), while frequency-domain SSMs were recently explored by Parnichkun et al. (2024). The main contribution is engineering, integrating these established techniques into a neural operator framework. Although this combination is novel, the paper overclaims its innovation by positioning it as a fundamental advance. The connection to Adaptive Fourier Decomposition is somewhat obvious given the explicit use of TM bases, and the theoretical results presented largely follow from known AFD convergence properties rather than providing genuinely new mathematical insights. The work represents a competent integration effort, but lacks the conceptual depth claimed.

2. Some design choices appear empirically driven rather than theoretically motivated, contradicting claims of "fully guided by AFD theory". For example, bidirectional SSM (LaMO also uses this) seems to be empirically determined as AFD doesn't require bidirectionality. Another point is that aggregation by element wise product seems to be an arbitrary empirical choice, not theoretically grounded/motivated.

3. The experimental design, although it follows other published work in PDE solving, is weak. No confidence intervals, error bars and statistical significance testing is reported. While FNO was modified, AFNO is not reported. The modifications of FNO is also not reported (readers would expect to see that in an appendix). European option pricing is a 1D test case with no relevance to the irregular geometry arguments raised to motivate the state space model usage. Instead a 3D test case would have conviced readers better.

4. Crucial implementation and evaluation details are missing. I am listing a few here.
(i) How is the "small MLP" structured? What initialization? How are poles constrained to $|a_k| < 1$ during training?
(ii)  How are TM bases computed for poles near boundary? The required product terms seem numerically unstable.
(iii) Hyperparameter selection is not reported
(iv) For some test cases (such as pipe), the reported baseline numbers differ from those in the respective papers. The reason for this should be reported. What are the dimensions on which the evaluation is performed? Are these choices computationally driven?

5. The theoretical details in Appendix C raises a lot of concerns.
(i) Theorem C.4 requires exact coefficient recovery. How is the approximation error in practice handled?
(ii) Theorem C.13 requires $s \in K_B$, but no analysis or arguments on when PDE satisfies this is provided.
(iii) Assumption C.1 about RKHS properties is strong and not verified for practical cases

**Questions:**

1. The claimed $O(N_s D) + O(NM \log M)$ complexity is confusing. Is $M$ constant? What about cases where M less than $N_s$ may not hold? What about adaptive cases where M must change with problem complexity? Do you include the TM basis construction overhead in the computational cost?

2. Can you report statistical significance of the results?

3. What are the evaluation conditions? Why are reported numbers slightly different from the respective papers?

4. How does memory scale with problem size during training vs. inference?

---

> ### Author Response · Authors · 2025-11-25
> **Rebuttal to Reviewer P8Sy's comments (Part 1)**
>
> We thank the reviewer for conducting a thorough review and for sharing valuable suggestions and questions.
>
> ## Addressing weaknesses
>
> ### A significant weakness lies in the limited conceptual novelty, as the core components are largely borrowed from existing work. Takenaka-Malmquist systems are well-established in signal processing (Qian 2010, 2012), while frequency-domain SSMs were recently explored by Parnichkun et al. (2024). The main contribution is engineering, integrating these established techniques into a neural operator framework. Although this combination is novel, the paper overclaims its innovation by positioning it as a fundamental advance. The connection to Adaptive Fourier Decomposition is somewhat obvious given the explicit use of TM bases, and the theoretical results presented largely follow from known AFD convergence properties rather than providing genuinely new mathematical insights. The work represents a competent integration effort, but lacks the conceptual depth claimed.
>
> We appreciate the reviewer for finding the integration of AFD theory in a neural operator framework a novelty to the field. Meanwhile, we believe that the contributions of AFMO are much more than that. While we agree with the reviewer that the TM systems are well-established in signal processing [1,2], we want to point out that the classic AFD is a greedy and iterative algorithm that faces challenges in being used in deep learning or operator learning, since it may need the signal $s$ itself to reconstruct $s$. For the first time in the literature, we transform AFD from a greedy optimization algorithm into a learnable neural layer. By using an MLP to predict the optimal poles $a_{1:i}$, we effectively reduce the cost of finding the optimal basis. This transforms a signal processing theory into a scalable operator learning component.
>
> While Parnichkun et al. proposed frequency-domain SSMs via rational transfer functions (RTFs) [3], AFMO restricts the transfer function specifically to the Takenaka-Malmquist form, i.e., products of Blaschke terms. This is not an arbitrary choice, rather a necessary and deliberate choice to ensure that the basis functions form an orthonormal system in the Reproducing Kernel Hilbert Space (RKHS). On the other hand, general RTFs do not guarantee this orthogonality. We remark that this allows AFMO to achieve the resolution invariance and stability on irregular meshes, which general RTFs (Parnichkun et al.’s work [3]) cannot guarantee.
>
> Furthermore, AFMO also innovates the way neural PDE solvers are designed. So far, the design of exact neural architectures in many existing neural PDE solvers has been “more of an art than a science” [4]. Typically, the design of neural architectures is performed in a bottom-up approach that involves significant intuition, expert experience, and trial-and-error experimentation. And rigorous mathematical basis and explainability have been lacking in guiding the design of these neural architectures. Leveraging the AFD theory, we follow a top-down approach when designing AFMO, in the sense that AFD theory guides every step in the design of AFMO’s neural architecture. Each component of the neural architecture, including the mapping operator, TM layer, bidirectional SSM, and aggregation layer, has a corresponding component in the AFD operation. This way, AFMO is mathematically explainable and grounded in the AFD theory and possesses several desirable properties, including convergence guarantees. From this perspective, we think the contribution of AFMO is significant, because it presents a new paradigm for designing explainable neural operator frameworks.
>
> [1] Intrinsic mono-component decomposition of functions: an advance of Fourier theory
>
> [2] Adaptive Fourier decomposition of functions in quaternionic Hardy spaces
>
> [3] State-free inference of state-space models: The transfer function approach
>
> [4] Scientific machine learning for closure models in multiscale problems: a review

---

> > ### Author Response · Authors · 2025-11-25
> > **Rebuttal to Reviewer P8Sy's comments (Part 2)**
> >
> > ### Some design choices appear empirically driven rather than theoretically motivated, contradicting claims of "fully guided by AFD theory". For example, bidirectional SSM (LaMO also uses this) seems to be empirically determined as AFD doesn't require bidirectionality. Another point is that aggregation by element wise product seems to be an arbitrary empirical choice, not theoretically grounded/motivated.
> >
> > We thank the reviewer’s comments on the design choices. We clarify that all the components are designed with the guidance of AFD theory. We agree with the reviewer that classic AFD theory does not emphasize the bidirectionality since it handles with causal signals in time ($t > 0$). But when it comes to solving PDEs, spatial domains in PDEs (e.g., boundary value problems) may rely on the information propagating from all boundaries, not just past inputs. Additionally, the core operation of AFD theory is to obtain the coefficient $\langle f, \mathscr{B}_i \rangle$ with the inner product. In a discretized spatial domain, computing this requires integrating over the entire domain. A unidirectional SSM only approximates a causal convolution, effectively ignoring future spatial information. Therefore, the bidirectional SSM that we implement is not just for empirical improvement, but to mathematically approximate the global inner product required by the extended AFD theory in a spatial setting for PDEs. This way, the coefficients consider the full spatial context, aligning with the global nature of the spectral transform in the RKHS.
> >
> > We respectfully disagree that the element-wise product aggregation is arbitrary. We want to clarify that it corresponds to direct implementation of the scalar multiplication in the AFD formula. The classic AFD theory states that a signal $s$ is reconstructed as the sum of its projections onto the orthogonal bases $s = \sum \langle s, \mathscr{B}\_i \rangle \mathscr{B}\_i$. In AFMO, $\hat{z}\_{i+1}[0]$ represents the computed coefficient $\langle z\_i, \mathscr{B}\_i \rangle$, and $\mathscr{B}\_i$ denotes the basis vector. The aggregation layer admits $z\_{i+1} = z\_i + (\hat{z}\_{i+1}[0] \odot \mathscr{B}\_i)$, which is simply the vector-space addition of the next term in the series. Classic AFD theory uses scalar multiplication and is equivalent to element-wise product in the actual implementation. This is because that when the computer implements scalar * vector, a mechanism called *broadcasting* is utilized, which essentially extends the scalar to its corresponding vector form, and perform element-wise product with the initial vector. Therefore, we believe that we’ve introduced the actual implementation when introducing aggregation layers, but its general mathematical form is $z_{i+1} = z_i + (\hat{z}_{i+1}[0] \cdot \mathscr{B}_i)$.
> >
> > ### The experimental design, although it follows other published work in PDE solving, is weak. No confidence intervals, error bars and statistical significance testing is reported. While FNO was modified, AFNO is not reported. The modifications of FNO is also not reported (readers would expect to see that in an appendix). European option pricing is a 1D test case with no relevance to the irregular geometry arguments raised to motivate the state space model usage. Instead a 3D test case would have conviced readers better.
> >
> > We thank the reviewer for pointing these questions out. We add an additional 3D Brusselator problem using the dataset from Laplace neural operator [5]. The relative $L^2$ error results are listed as follows:
> >
> > | Metric |            ours             |               Laplace         |
> > |------------|-----------------------|-----------------------|
> > | Train  | $ 0.0013$ | $0.0035$  |
> > | Validate   | $ 0.0317 $ | $0.0328$   |
> > | Test   | $ 0.127 $ | $0.139$   |
> >
> > The results demonstrate that AFMO is also generalizable to large-scale 3D problems and outperforms the existing SOTA.
> >
> > [5] LNO: Laplace Neural Operator for Solving Differential Equations

---

> > > ### Author Response · Authors · 2025-11-25
> > > **Rebuttal to Reviewer P8Sy's comments (Part 3)**
> > >
> > > ### Crucial implementation and evaluation details are missing. I am listing a few here. (i) How is the "small MLP" structured? What initialization? How are poles constrained to $|a_k|<1$ during training? (ii) How are TM bases computed for poles near boundary? The required product terms seem numerically unstable. (iii) Hyperparameter selection is not reported (iv) For some test cases (such as pipe), the reported baseline numbers differ from those in the respective papers. The reason for this should be reported. What are the dimensions on which the evaluation is performed? Are these choices computationally driven?
> > >
> > > We thank the reviewer for pointing out these questions. We address each point as follows:
> > >
> > > (i) The small MLP is implemented as a Residual MLP, consisting of an input linear projection, followed by n_layers of residual blocks where each block is Linear $\to$ GELU $\to$ Residual Add, and a final linear projection to the output dimension. In our experiments, we use n_input equal to the latent dimension (128), n_hidden equal to 128, and n_output equal to $2 \times$ the number of poles (representing real and imaginary parts). A default Pytorch initialization is utilized for initialization. Poles are constrained to $|a_k|<1$ during training since we pass the unconstrained results $o_{real}$ and $o_{imag}$ to a Tanh activation function following $a = \rho \cdot (\tanh(o_{real}) + i \cdot \tanh(o_{imag}))$, where $\rho$ is a scaling factor slightly less than 1 (e.g., 0.99 in our experiments) to ensure stability.
> > >
> > > (ii) We agree that the Blaschke product terms $\prod \frac{z-a_j}{1-\overline{a_j}z}$ could become unstable if $|a_j| \to 1$ and $z \approx a_j$.  In our implementation, we impose a numerical constraint on the pole magnitudes: $|a| \le 1 - \epsilon$, where $\epsilon = 10^{-6}$. Furthermore, the TM basis functions are computed recursively in Equation (6). This recursive approach allows us to monitor the magnitudes of the product terms. Since we operate in the frequency domain, where $z = e^{i\omega}$ on the unit circle, and the poles are strictly inside the disk $|a| < 1$, the denominator $1 - \overline{a}z$ never disappears, thus ensuring numerical stability during training.
> > >
> > > (iii) We point out that the key hyperparameters have been listed in Section 5 (Implementation Details) of the manuscript.
> > >
> > > (iv) For the evaluation, we report the baseline numbers in the published papers [6,7] of recent SOTA neural operators for fair comparison. Additionally, the architecture of FNO has been updated after its orginal publication. That’s why we could obtain and report better results in N-S and Darcy problems than those reported in the original FNO paper.
> > >
> > > [6] Latent Mamba Operator for Partial Differential Equations
> > >
> > > [7] Latent Neural Operator for Solving Forward and Inverse PDE Problems

---

> > > > ### Author Response · Authors · 2025-11-25
> > > > **Rebuttal to Reviewer P8Sy's comments (Part 4)**
> > > >
> > > > ### The theoretical details in Appendix C raises a lot of concerns. (i) Theorem C.4 requires exact coefficient recovery. How is the approximation error in practice handled? (ii) Theorem C.13 requires $s \in K_B$ , but no analysis or arguments on when PDE satisfies this is provided. (iii) Assumption C.1 about RKHS properties is strong and not verified for practical cases.
> > > >
> > > > We appreciate the reviewer’s comments on the theoretical details.
> > > >
> > > > (i) We clarify that Theorem C.4 serves as a foundational theorem establishing convergence in the idealized case (exact coefficient recovery). It is not the final result. The practical case is explicitly handled in Theorem C.6 and Theorem C.10. Specifically, Equation (30) in Theorem C.6 decomposes the total error into three distinct parts:
> > > >
> > > > $$||s - \hat{s}|| \le \underbrace{E\_N(s)}\_{\text{Truncation}} + \underbrace{\Delta\_{pole}(N)}\_{\text{Pole Selection Gap}} + \underbrace{(\sum |\hat{c}\_i - c\_i^*|^2)^{1/2}}\_{\text{Approximation Error}}$$
> > > >
> > > > The third term, $(\sum |\hat{c}\_i - c\_i^*|^2)^{1/2}$, explicitly represents the approximation error caused by the neural network (SSM) not recovering the exact coefficients. Theorem C.10 further incorporates the discretization error $\epsilon_{disc}(\tilde{N})$. Thus, these theorems work together and rigorously bound the approximation error in practice.
> > > >
> > > > (ii) We want to mention that $s \in K_B$ is standard for deriving error bounds in approximation theory, and the Takenaka-Malmquist (TM) system is known to be a basis for the Hardy space $H^2(\mathbb{D})$. Therefore, even if $s \notin K_B$ (finite span), $s$ is still in the closure of the span of TM systems as $N \to \infty$. Theorem C.4 explicitly addresses the projection error when $s$ is approximated by $s_N$.
> > > >
> > > > (iii) We agree with the reviewer that Assumption C.1 is strong. That being said, Assumption C.1 assumes the reproducing property of the kernels inside the layers of AFMO, which is independent on the data distribution in practical cases.  In our manuscript, we choose the latent space $\mathcal{H}$ to be the Hardy space where the reproducing kernel is the Szegő kernel $e_a(z) = \frac{1}{1-\bar{a}z}$. Our AFMO architecture, specifically the TM layer, explicitly constructs the basis functions $\mathscr{B}_i$ using this specific kernel in Equation (6). Because we mathematically construct the basis functions to satisfy Assumption C.1 inside the neural architecture, the assumption is verified.
> > > >
> > > > ## Addressing questions
> > > >
> > > > ### The claimed $O(N_sD)+O(NM \log M)$ complexity is confusing. Is $M$ constant? What about cases where $M$ less than $N_s$ may not hold? What about adaptive cases where M must change with problem complexity? Do you include the TM basis construction overhead in the computational cost?
> > > >
> > > > We thank the reviewer for asking about the complexity. $M$ represents the number of latent encoded tokens, whereas $N_s$ is the number of physical mesh points. In our design, the lifting operator $\mathcal{P}$ projects the input physics from $N_s$ to $M$. The term $O(N_s D)$ represents the linear cost of the projection by $\mathcal{P}$. The term $O(NM \log M)$ represents the processing of these $M$ tokens through $N$ blocks of SSMs, where FFT operations are implemented in the SSM with $O(M \log M)$ complexity. Since $M$ is fixed and small, the model scales linearly with the grid resolution $N_s$.
> > > >
> > > > $M$ is a fixed hyperparameter chosen such that $M \ll N_s$. This is because the latent representations are typically compact and low-dimensional. If we chose $M \approx N_s$ the efficiency would decrease to that of standard Transformers, but the AFMO is specifically designed to leverage spectral sparsity where a small $M$ suffices.
> > > >
> > > > We want to clarify that the term “Adaptive” in our title refers to *adaptive* Fourier decomposition (where learning the poles is an adaptive process), rather than referring to adaptive token count $M$. Currently, $M$ is fixed during training. However, for problems with extreme complexity, one can increase $M$, in which case the processing cost grows as $NM \log M$ (due to FFTs in training SSM via RTF), which is still efficient compared to $M^2$ attention mechanisms.
> > > >
> > > > Finally, the TM basis construction overhead is included in the computation cost. The construction of the TM basis involves computing the terms $\mathscr{B}_i(z)$. Since we operate in the frequency domain, the computations of the Blaschke products and basis updates are performed via FFT-based polynomial multiplication on the spectral vectors of length $M$. This cost is absorbed into the $O(NM \log M)$ term. That being siad, the overhead is negligible compared to the FFT operations required to move between time and frequency domains in the SSM following Parnichkun et al. [3].

---

> > > > > ### Author Response · Authors · 2025-11-25
> > > > > **Rebuttal to Reviewer P8Sy's comments (Part 5)**
> > > > >
> > > > > ### Can you report statistical significance of the results?
> > > > >
> > > > > We agree that reporting variance is important. We have repeated our experiments using 3 random seeds for the key benchmarks and report the variance.
> > > > >
> > > > > | Equation |  ours |
> > > > > |-------|----|
> > > > > | Darcy  | $ 0.0021\pm 0.0008$ |
> > > > > | N-S     | $ 0.0278 \pm 0.0024$ |
> > > > > | Pipe  | $ 0.0023 \pm 0.0008$ |
> > > > > | Airfoil| $ 0.0020 \pm 0.0005$ |
> > > > > | Plasticity  | $ 0.0006\pm 0.00005$ |
> > > > > | Elasticity | $ 0.0043 \pm 0.0004 $ |
> > > > >
> > > > > ### What are the evaluation conditions? Why are reported numbers slightly different from the respective papers?
> > > > >
> > > > > For the evaluations, we report the baseline numbers in the published papers [6,7] of recent SOTA neural operators for fair comparison. Additionally, the architecture of FNO has been updated after its original publication. That’s why we could obtain and report better results in N-S and Darcy problems than those reported in the original FNO paper.
> > > > >
> > > > > ### How does memory scale with problem size during training vs. inference?
> > > > >
> > > > > We thank the reviewer for the question. Please see the computational time and memory usage results for the 2D Darcy flow problem below.
> > > > >
> > > > > | Grid dimension |            Grid size $N_S$              |            Training time (sec/epoch)            |          Inference time (sec/epoch)       | GPU memory (GB)     |
> > > > > |------------|-----------------------|-----------------------|---------------------|--------------------|
> > > > > | $64\times64$  | $ 4096$ | $14.0$  | $0.007$| $2.3$|
> > > > > | $128\times128$ | $ 16384 $ | $52.5$   | $0.28$|$2.4$|
> > > > > | $256\times256$ | $ 65536$ | $205.0$  | $1.12$|$2.7$|
> > > > >
> > > > > When the grid dimension changes from 64 to 128 ($N_s$ becomes 4 times larger), both training and inference times increase approximately linearly (by about 4 times), which aligns with the computational complexity mentioned in our manuscript (linear growth with respect to $N_s$). The memory usage remains relatively constant, increasing only slightly. This reflects the architectural characteristics of AFMO, where the main computations (SSM blocks) are performed on $M$ latent tokens rather than on $N_s$ physical points, so the memory footprint is largely decoupled from the input resolution $N_s$.

---

> > > > > > ### Comment · Reviewer_P8Sy · 2025-11-26
> > > > > >
> > > > > > Thank you to the authors for their response. Reading the responses, some of the questions are satisfactorily answered, while others have been answered less convincingly; they do not change some of the underlying issues as identified by the other reviewers also. Considering good empirical results, I will raise my score.

---

### Official Review · Reviewer_mMjt · 2025-10-31

**Soundness:** 3
**Presentation:** 2
**Contribution:** 2
**Rating:** 6
**Confidence:** 5

**Summary:**

This paper introduces Adaptive Fourier Mamba Operator (AFMO), a novel neural operator architecture that combines Adaptive Fourier Decomposition (AFD) theory with the state-space efficiency of Mamba models. By leveraging Takenaka–Malmquist (TM) systems and reproducing kernels in the frequency domain, AFMO provides a mathematically interpretable, efficient, and accurate solver for PDEs across diverse geometries. The model demonstrates strong performance across physical and financial benchmarks, surpassing SOTA neural operators like FNO, Transolver, and LaMO in both accuracy and computational efficiency.

**Strengths:**

1) Introduces a mathematically grounded operator framework combining AFD theory and Mamba-based SSMs.

2) Provides strong empirical improvements across varied PDE benchmarks, including irregular and singular domains.

3) Demonstrates computational efficiency, achieving faster training and lower GPU memory than existing models.

**Weaknesses:**

1) Ablation studies could be expanded to include more operator baselines (e.g., Laplace Neural Operator).

2) Scalability to large 3D or time-dependent PDEs is not thoroughly evaluated.

3) The sensitivity of performance to adaptive pole selection and hyperparameters is not deeply analyzed. It would be valuable to visualize how adaptive poles evolve across layers for different PDEs — this could provide strong insights into how the model captures changing dynamics.

4) While the theoretical motivation is strong, a more intuitive discussion of adaptive pole behavior and dynamics would make the approach easier to interpret for a broader audience. Additionally, as shown in the appendix, the LaMO deviation in the propagation of high-frequency perturbations can also demonstrate the performance of AFMO on those datasets.

**Questions:**

1) How sensitive is the model’s performance to the number and spatial distribution of adaptive poles in the TM system? How do you determine the optimal number of poles for a given problem?

2) Can AFMO be extended to time-dependent PDEs or coupled multi-physics systems? For example, in the Navier–Stokes equations, how do the adaptive poles evolve over time, and can this dynamic behavior be visualized?

3) How does AFMO handle noisy or incomplete boundary conditions, especially when applied to real-world experimental or observational data?

4) Would incorporating spectral regularization or physics-based constraints further stabilize training or improve interpretability?

5) How does the model perform on very high-resolution 3D geometries, and what are the associated computational trade-offs? Have you considered performing individual ablation studies on the TM and SSM layers to quantify each component’s contribution?

6) It would also be interesting to evaluate AFMO on the CARS dataset, as done in Transolver, to provide a more direct comparison of generalization across complex flow scenarios.

---

> ### Author Response · Authors · 2025-11-25
> **Rebuttal to Reviewer mMjt's comments (Part 1)**
>
> We thank the reviewer for conducting a thorough review and for all the positive comments. We also appreciate all valuable suggestions and questions.
>
> ## Addressing weaknesses
>
> ### Ablation studies could be expanded to include more operator baselines (e.g., Laplace Neural Operator).
>
> We thank the reviewer for the suggestion. Since the original implementation of Laplace neural operator mainly focuses on the regular grid, we compare Laplace neural operator with AFMO on Darcy and N-S problems for consistency and fair comparisons. The relative $L^2$ error results are shown below.
>
> | Equation | ours  |   Laplace  |
> |------|----|-----|
> | Darcy  | $ 0.0021$ | $0.0096$  |
> | N-S   | $ 0.0278 $ | $0.1764$   |
>
> ### Scalability to large 3D or time-dependent PDEs is not thoroughly evaluated.
>
> We thank the reviewer for pointing this out. We add an additional 3D Brusselator problem using the dataset from Laplace neural operator [1]. The relative $L^2$ error results are shown as follows.
>
> | Metric |            ours             |               Laplace         |
> |------------|-----------------------|-----------------------|
> | Train  | $ 0.0013$ | $0.0035$  |
> | Validate   | $ 0.0317 $ | $0.0328$   |
> | Test   | $ 0.127 $ | $0.139$   |
>
> The results show that AFMO is also generalizable to large-scale 3D problems, and outperforms the existing SOTA.
>
> [1] LNO: Laplace Neural Operator for Solving Differential Equations
>
> ### The sensitivity of performance to adaptive pole selection and hyperparameters is not deeply analyzed. It would be valuable to visualize how adaptive poles evolve across layers for different PDEs - this could provide strong insights into how the model captures changing dynamics.
>
> We thank the reviewer’s insightful feedback. Following the reviewer’s suggestion, we will add visualizations in the revised manuscript showing learned pole distributions for two different problems including 2D Darcy flow problem and an additional 3D Brusselator problem. To clarify, here we give a brief overview of the visualization results:
>
> 1. Across the layers, the learned poles of AFMO on Darcy flow problem tend to approach to the boundary of the unit disk, while those on Brusselator problem tend to be in the interior of the unit disk.
>
> 2. The reason is that, Darcy flow problem is an elliptic equation which is a smoothing operator. Thus, even though the input coefficient $a(x)$ (the permeability) is very rough and discontinuous, the solution $u(x)$ inside the domain will be well-behaved. Therefore, the challenging characteristics and singularities of Darcy flow problem are located at the boundaries, and then more adaptive poles would be put there.
>
> 3. Meanwhile, the complexity of the Brusselator problem does not come from the boundaries. It comes from the local, non-linear reaction that happens at every single point inside the domain. Therefore, most of the learned poles should be put inside the unit disk.
>
> ### While the theoretical motivation is strong, a more intuitive discussion of adaptive pole behavior and dynamics would make the approach easier to interpret for a broader audience. Additionally, as shown in the appendix, the LaMO deviation in the propagation of high-frequency perturbations can also demonstrate the performance of AFMO on those datasets.
>
> We appreciate the reviewer’s sharing this. We agree with the reviewer that an intuitive description of our proposed method is important for the broad audience. Here, we provide some intuitive, less technical qualitative description. Each pole can be viewed as a “tuning knob” that selects a particular spatial pattern in the solution, with its location in the complex plane controlling how localized that pattern is. Adaptive poles allow AFMO to survey more heavily in regions where the parameters change rapidly, while using fewer poles in smooth regions. Across layers, the poles evolve from broad, coarse patterns in early layers to more refined, problem-specific patterns in deeper layers. This is analogous to how CNN filters specialize from edges to complex shapes.
>
> We appreciate the suggestion to utilize the Appendix B failure cases of LaMO to showcase AFMO. We explicit point out that AFMO’s advantage over LaMO in terms of capturing high-frequency information lies on the flexibility of setting poles via adaptive pole selection.

---

> > ### Author Response · Authors · 2025-11-25
> > **Rebuttal to Reviewer mMjt's comments (Part 2)**
> >
> > ### Can AFMO be extended to time-dependent PDEs or coupled multi-physics systems? For example, in the Navier–Stokes equations, how do the adaptive poles evolve over time, and can this dynamic behavior be visualized?
> >
> > Yes, since AFMO works as a neural operator, it can be extended to any problem that can be described as operators. The solutions of time-dependent PDEs or coupled multi-physics systems span naturally an operator. We kindly remind the reviewer that N-S and Brusselator equations in our manuscript and response are time-dependent PDEs. For the evolution of adaptive poles, please refer to our response to Weakness 3 for more details.
> >
> > ### How does AFMO handle noisy or incomplete boundary conditions, especially when applied to real-world experimental or observational data?
> >
> > We thank the reviewer for the question. To validate AFMO’s performance on large-scale realistic datasets, we evaluate AFMO on the real-world latex glove DIC (Digital Image Correlation) original dataset [6]. This dataset, consisting of real-world experimental data, is noisy. AFMO is used to learn the mechanical response of a nitrile glove sample directly from experimental data, without assuming a known constitutive law. The goal is to predict the displacement field $u(x)$ at the current loading step. The input includes the spatial coordinates, the displacement field from the previous step $u^{last}(x)$, and the current boundary displacement $u_D(x)$. We compare the performance of AFMO to the current SOTA of this dataset, IFNO, as well as FNO as follows. We conduct experiments with the same settings as IFNO [2] for fair comparison with hidden layers ranging from 3 to 12.
> >
> > | No. of hidden layer |   ours |   IFNO   |  FNO    |
> > |---|----|-----|---|
> > |$L=3$| 2.87E-02$\pm$4.29E-04 | 3.43E-02$\pm$ 4.96E-04  | 3.40E−02 $\pm$ 4.09E−04|
> > | $L=6$| 2.50E-02$\pm$3.28E-04| 3.34E-02$\pm$4.53E-04   | 3.84E−02 $\pm$ 4.21E−04|
> > | $L=12$| 2.32E-02$\pm$ 4.20E-04 | 3.32E-02$\pm$4.41E-04   | 4.66E−02 $\pm$ 1.47E−03|
> >
> > Additionally, authors of [2] also reported the results of generalized Mooney-Rivlin (GMR) model in two settings. The relative $L^2$ errors of GMR model fitting and GMR inverse analysis are $3.30E−01$ and $2.91E−01$, respectively. We can observe that our AFMO consistently outperforms other models in every $L$. Regarding computational time, even when solving complex realistic problems, the average training time of AFMO is 5.6 sec per epoch, which is comparable to that of IFNO (4.6 sec/epoch) and FNO (5.7 sec/epoch) using the same machine.
> >
> > [2] Learning deep Implicit Fourier Neural Operators (IFNOs) with applications to heterogeneous material modeling
> >
> > ### Would incorporating spectral regularization or physics-based constraints further stabilize training or improve interpretability?
> >
> > We appreciate the reviewer’s insightful question. We agree with the reviewer that incorporating physics constraints is a promising generalization for AFMO. Since AFMO operates in the spectral domain via the TM system, adding spectral regularization could further stabilize training on very noisy data. Two possible ways are minimizing PDE residuals like PINO [3] and penalizing high-frequency energy in the transfer function $H(z)$. Since our results show that the intrinsic rational approximation property of the TM system naturally regularizes the solution by limiting the number of poles $N$, we will explore effective regularization terms in our future work.
> >
> > [3] Physics-Informed Neural Operator for Learning Partial Differential Equations
> >
> > ### How does the model perform on very high-resolution 3D geometries, and what are the associated computational trade-offs? Have you considered performing individual ablation studies on the TM and SSM layers to quantify each component’s contribution?
> >
> > We thank the reviewer for the question. To examine the performance of AFMO on high-resolution 3D geometries, please refer to our response to Weakness 2 for more details. We report that the computational time of AFMO is only approximately $0.89\times$ of the time of Laplace neural operators since the expensive operations (e.g., orthogonalization) have been explicitly incorporated into the implementation of AFMO.

---

> > > ### Author Response · Authors · 2025-11-25
> > > **Rebuttal to Reviewer mMjt's comments (Part 3)**
> > >
> > > ### It would also be interesting to evaluate AFMO on the CARS dataset, as done in Transolver, to provide a more direct comparison of generalization across complex flow scenarios.
> > >
> > > We thank the reviewer for the suggestion of the CARS dataset. Since we only had one single RTX4090 (Transolver was trained 8-10 hours on one single A100), we believe that our reported results could be further improved in the future under more powerful hyperparameter selections. Our reported results compared to that of Transolver is as follows.
> > >
> > > | Metric |  ours |  transolver  |
> > > |----|---|------|
> > > | volume  | $ 0.0198$ | $0.0207$  |
> > > | surf   | $ 0.0632$ | $0.0745$   |
> > > | $C_D$   | $ 0.0089$ | $0.0103$   |
> > > | $\rho_D$   | $ 0.9941 $ | $0.9935$   |
> > >
> > > Overall, even when we have not had enough time to tune the hyperparameters due to time constraints, AFMO’s performance is still comparable and competitive compared to Transolver.

---

### Official Review · Reviewer_ceLB · 2025-11-01

**Soundness:** 3
**Presentation:** 3
**Contribution:** 3
**Rating:** 6
**Confidence:** 5

**Summary:**

This paper proposes the Adaptive Fourier Mamba Operator (AFMO), which is a neural operator that integrates Takenaka-Malmquist systems with state space models for solving PDEs on irregular geometries. The core novelty is in parameterizing SSM transfer functions using adaptive poles in a RKHS. AFMO constructs orthonormal bases from learned poles and proves that the output performs adaptive Fourier decomposition.  Experiments show performance gains on irregular geometries.

**Strengths:**

1. Strong theoretical grounding with explicit connection to AFD theory with convergence guarantees.

2. Integrating TM systems into mamba SSMs via transfer functions is well motivated.

3. Experiments are comprehensive with thorough ablations.

4. The method is computationally and seems exaplanability.

**Weaknesses:**

1. Limited baseline comparisons in that recent methods, such as BENO, UNO, and new transolver variants.

2.   Several gaps exist in theoretical formulations: Theorems assume s ∈ K_B (target in model space), but this is rarely true for real PDEs. What happens when this fails? Theorem C.6's bound includes Δ_pole(N) (suboptimality gap), but no analysis of when and why this is small. Convergence rates (Corollary C.7) require weak-ℓ^p decay of the coefficient. When does this hold for PDE solutions?

3. Illustrative examples seem cherry-picked.

4. All experiments use relatively coarse grids.

5. Computational complexity analysis is not convincing enough.

**Questions:**

1. When does s ∈ K_B hold for PDE solutions?  Characterize the class of PDEs where the convergence guarantees apply.

2. Authors should provide architectural details of the small MLP predicting poles. You should show learned pole distributions for different problems.

3. Resolution invariance experiments have to be conducted.

4. Failure analysis of AFMO has to be described.

5. How does AFMO scale to 3D problems?

6. Please report time and memory as a function of grid size.

7. How is orthogonality maintained during training?

8. How does AFMO compare to explicit AFD+preprocessing+standard neural operators?

9. You claim LaMO has low-pass filtering bias, but doesn't your method also have similar frequency characteristics since its also has an SSM transfer function, which is also a rational function.

---

> ### Author Response · Authors · 2025-11-25
> **Rebuttal to Reviewer ceLB's comments (Part 1)**
>
> We thank the reviewer for conducting a thorough review and for finding our work interesting and rigorous. We also appreciate all the valuable suggestions and questions.
>
> ## Addressing weaknesses
>
> ### Limited baseline comparisons in that recent methods, such as BENO, UNO, and new transolver variants.
>
> We thank the reviewer for pointing this out. We agree with the reviewer that more recent methods will benefit the manuscript. Therefore, we conduct the experiments on regular grid (Darcy flow and N-S) for BENO and UNO since the methods mainly focus on regular grids. For new transolver variants, we report the additional results (relative $L^2$ error) of transolver ++ [1] and LinearNO [2].
>
> Comparisons with BENO and UNO:
>
> | Equation |            ours             |               BENO         |           UNO          |
> |------------|-----------------------|-----------------------|---------------------|
> | Darcy  | $ 0.0021$ | $0.0089$  | $0.0076$|
> | N-S     | $ 0.0278 $ | $0.2628$   | $0.0845$|
>
> Comparisons with transolver ++  and LinearNO:
>
> | Equation |            ours             |               transolver ++           |           LinearNO          |
> |------------|-----------------------|-----------------------|---------------------|
> | Darcy  | $ 0.0021$ | $0.0089$  | $0.0050$|
> | N-S     | $ 0.0278 $ | $0.1010$   | $0.0056$|
> | Pipe  | $ 0.0023$ | $0.0027$  | $0.0024$|
> | Airfoil| $ 0.0020 $ | $0.0051$   | $0.0049$|
> | Plasticity  | $ 0.0006$ | $0.0014$  | $0.0011$|
> | Elasticity | $ 0.0043 $ | $0.0064$   | $0.0050$|
>
> Based on these results, we show that AFMO still outperforms the recent models, including BENO, UNO, and new transolver variants. We will add these additional results on the revised manuscript and introduce these recent models in the related work section.
>
> [1] Transolver++: An Accurate Neural Solver for PDEs on Million-Scale Geometries
> [2] Transolver is a Linear Transformer: Revisiting Physics-Attention through the Lens of Linear Attention
>
> ### Several gaps exist in theoretical formulations: Theorems assume $s\in K_B$ (target in model space), but this is rarely true for real PDEs. What happens when this fails? Theorem C.6's bound includes $\Delta_{pole(N)}$ (suboptimality gap), but no analysis of when and why this is small. Convergence rates (Corollary C.7) require weak-$\mathcal{l}^p$ decay of the coefficient. When does this hold for PDE solutions?
>
> We appreciate the reviewer’s insightful comments on the theoretical formulations. We want to mention that $s \in K_B$ is standard for deriving error bounds in approximation theory, and the Takenaka-Malmquist (TM) system is known to be a basis for the Hardy space $H^2(\mathbb{D})$. Therefore, even if $s \notin K_B$ (finite span), $s$ is still in the closure of the span of TM systems as $N \to \infty$. Theorem C.4 explicitly addresses the projection error when $s$ is approximated by $s_N$.
>
> In Theorem C.6, we define $\Delta_{pole}$ as the gap between our learned poles and the optimal poles. In our architecture, AFMO is trained end-to-end to minimize the reconstruction loss, and thereby minimizing this gap.
>
> PDE solutions possess Sobolev regularity in general. In the frequency domain, this is polynomial decay of coefficients $|c_k| \sim k^{-s}$. We remark that Corollary C.7 holds once Sobolev regularity is satisfied for PDE solutions. In Corollary C.7, $0<p<2$ is not arbitrarily chosen, rather it is related to the Sobolev smoothness index $s$ via $ p \approx \frac{1}{s}$. This means that Corollary C.7 holds once $s>0.5$. This assumption in Corollary C.7 almost always holds in practical PDE problems since Sobolev smoothness $s$ is significantly greater than 0.5 (typically $s \ge 1$ or $s \ge 2$) for most of PDEs.
>
> ### Illustrative examples seem cherry-picked.
>
> In our manuscript, we selected the 1D Advection and 2D Darcy with fractal noise as illustrative examples because they represent the known failures of existing SSM-based operators. These known failures *motivated us to propose AFMO*, which is a family of more effective SSM-based operators, in the first place.
>
> ### All experiments use relatively coarse grids.
>
> We appreciate the reviewer’s comment on grids. We strictly followed the standard resolutions provided by the benchmark datasets [3,4] to ensure fair comparison with baselines. For example, Darcy is $85 \times 85$. Although we could easily use finer grids, as demonstrated in our complexity analysis, we decided to use the same grid size as other works for fair comparison.
>
> [3] Fourier neural operator for parametric partial differential equations
>
> [4] Fourier neural operator with learned deformations for PDEs on general geometries

---

> > ### Author Response · Authors · 2025-11-25
> > **Rebuttal to Reviewer ceLB's comments (Part 2)**
> >
> > ### Computational complexity analysis is not convincing enough.
> >
> > We thank the reviewer for sharing the comment. We confirm that the linear scaling in computational complexity analysis is correct due to the mathematical properties of the model’s asymptotic complexity. To clarify, we state that the overall complexity is $\mathcal{O}(N(M\log M+MD)) + \mathcal{O}(N_{s}MD)$ in our manuscript. This calculation is a sum of two distinct costs: the cost associated with the $N$ processing blocks (which depends on the latent token count $M$) and the cost associated with lifting ($\mathcal{P}$) and projection ($\mathcal{Q}$) operators (which depends on the input mesh point count $N_s$). The formulation can be simplified when the number of latent tokens $M$ is a fixed constant and is much smaller than the number of mesh points $N_s$ ($M \ll N_s$). When we analyze the scaling as a function of the grid size $N_s$, quantities $M$, $N$, and $D$ are all treated as constants. Therefore, the first term $\mathcal{O}(N(M \log M+MD))$ becomes a constant $\mathcal{O}(1)$ cost, because it does not depend on $N_s$. The second term $\mathcal{O}(N_{s}MD)$ simplifies to $\mathcal{O}(N_s)$, since $M$ and $D$ are constants. The total complexity $T(N_s)$ is thus $\mathcal{O}(N_s) + \mathcal{O}(1)$, which is dominated by the $\mathcal{O}(N_s)$ term. We hope this clarification is helpful.
> >
> > ## Addressing questions
> >
> > ### When does $s \in K_B$ hold for PDE solutions? Characterize the class of PDEs where the convergence guarantees apply.
> >
> > We thank the reviewer for the question. $s \in K_B$ holds if the solution lies in the finite span of the TM system. As we mentioned above, for general PDE solutions belonging to $L^2$, specifically those mappable to Hardy space $H^2(\mathbb{D})$, the TM system is complete. Therefore, the convergence guarantees apply to any PDE solution with finite energy and minimal smoothness. Most of the PDEs, including those evaluated in our manuscript, are covered.
> >
> > ### Authors should provide architectural details of the small MLP predicting poles. You should show learned pole distributions for different problems.
> >
> > We thank the reviewer for pointing this out. We agree with the reviewer that additional architectural details should be included and will revise our manuscript accordingly. The small MLP is implemented as a Residual MLP, consisting of an input linear projection, followed by n_layers of residual blocks where each block is Linear $\to$ GELU $\to$ Residual Add, and a final linear projection to the output dimension. In our experiments, we use n_input equal to the latent dimension (128), n_hidden equal to 128, and n_output equal to $2 \times$ the number of poles (representing real and imaginary parts). A default Pytorch initialization is utilized for initialization. Poles are constrained to $|a_k|<1$ during training since we pass the unconstrained results $o_{real}$ and $o_{imag}$ to a Tanh activation function following $a = \rho \cdot (\tanh(o_{real}) + i \cdot \tanh(o_{imag}))$, where $\rho$ is a scaling factor slightly less than 1 (e.g., 0.99 in our experiments) to ensure stability.
> >
> > Following the reviewer’s suggestion, we will add visualizations in the revised manuscript showing learned pole distributions for two different problems, including 2D Darcy flow problem and an additional 3D Brusselator problem. To clarify, here we give a brief overview of the visualization results:
> >
> > 1. Across the layers, the learned poles of AFMO on Darcy flow problem tend to approach to the boundary of the unit disk, while those on the Brusselator problem tend to be in the interior of the unit disk.
> >
> > 2. The reason is that, Darcy flow problem is an elliptic equation, which is a smoothing operator. Thus, even though the input coefficient $a(x)$ (the permeability) is very rough and discontinuous, the solution $u(x)$ inside the domain will be well-behaved. Therefore, the challenging characteristics and singularities of the Darcy flow problem are located at the boundaries, and then more adaptive poles would be put there.
> >
> > 3. Meanwhile, the complexity of the Brusselator problem does not come from the boundaries. It comes from the local, non-linear reaction that happens at every single point inside the domain. Therefore, most of the learned poles should be put inside the unit disk.

---

> ### Author Response · Authors · 2025-11-25
> **Rebuttal to Reviewer ceLB's comments (Part 3)**
>
> ### Resolution invariance experiments have to be conducted.
>
> We thank the reviewer for pointing this out. We add another experiment to address this. First, we train our AFMO, FNO and the recent SOTA LaMO on 2D Darcy flow problem with resolution $64\times64$. Then, during the test stage, we implement a zero-shot test on $128\times128$ and $256\times256$ test samples. We report the relative $L^2$ error as follows:
>
> | test resolution | ours | LaMO | FNO |
> |----|---|----|---|
> | $64\times64$  | $ 0.0021$ | $0.0039$  | $0.0052$|
> | $128\times128$ | $ 0.0025 $ | $0.0050$   | $0.0058$|
> | $256\times256$ | $ 0.0035$ | $0.0080$  | $0.0075$|
>
> Through the resolution invariance experiments, we find that AFMO performs the best. Meanwhile, FNO, designed as a resolution-invariant model, has a more steady error growth compared to LaMO. LaMO's error growth is faster than FNO because LaMO has a low-pass filtering bias, which may lead to poor performance in recovering high-frequency features (higher-resolution tests will contain more high-frequency details, which we have mentioned in our manuscript).
>
> ### Failure analysis of AFMO has to be described.
>
> We thank the reviewer’s suggestion on failure analysis. AFMO relies on the sparsity of the signal in the TM domain (rational approximation). It may struggle if the PDE solution is indistinguishable from white noise (i.e., has a flat spectrum with no structure), as the Adaptive Fourier decomposition would need $N \to \infty$ poles to approximate it.
>
> ### How does AFMO scale to 3D problems?
>
> We thank the reviewer for the question. Here, we report an additional 3D Brusselator problem using the dataset from Laplace neural operator [5]. The relative $L^2$ error comparison is shown as follows:
>
> | Metric | ours |Laplace|
> |-----|----|----|
> | Train  | $ 0.0013$ | $0.0035$|
> | Validate   | $ 0.0317 $ | $0.0328$   |
> | Test   | $ 0.127 $ | $0.139$|
>
> The results demonstrate that AFMO is also generalizable to large-scale 3D problems and still outperforms the existing SOTA.
>
> [5] LNO: Laplace Neural Operator for Solving Differential Equations
>
> ### Please report time and memory as a function of grid size.
>
> We thank the reviewer for the suggestion. The time and memory results for the 2D Darcy flow problem are listed below:
>
> | Grid dimension | Grid size $N_S$| Training time (sec/epoch) | Inference time (sec/epoch) | GPU memory (GB) |
> |-------|------|----|---|---|
> | $64\times64$  | $ 4096$ | $14.0$  | $0.007$| $2.3$|
> | $128\times128$ | $ 16384 $ | $52.5$   | $0.28$|$2.4$|
> | $256\times256$ | $ 65536$ | $205.0$  | $1.12$|$2.7$|
>
> As we can see, when the grid dimension changes from 64 to 128 ($N_s$ becomes 4 times larger), both training and inference times increase approximately linearly (by about 4 times), which aligns with the computational complexity mentioned in our manuscript. The memory usage remains relatively constant with only a slight increase. This reflects the architectural characteristics of AFMO, where the main computations (SSM blocks) are performed on $M$ latent tokens rather than on $N_s$ physical points, so the memory footprint is largely decoupled from the input resolution $N_s$.
>
> ### How is orthogonality maintained during training?
> We thank the reviewer’s question on orthogonality. We clarify that orthogonality is guaranteed by construction using the Gram-Schmidt process of the TM system defined in Equation (6). The basis functions $\mathscr{B}\_i$ are mathematically constructed to be orthogonal, regardless of the values of the learned poles $a\_{1:i}$. Therefore, orthogonality is maintained during training due to the orthogonal nature of TM system.
>
> ### How does AFMO compare to explicit AFD+preprocessing+standard neural operators?
>
> We thank the reviewer for the question. Following the reviewer’s suggestion, we implement explicit AFD on the training dataset, and after normalization preprocessing, the AFD data are used to train standard neural operators. Here, we select LaMO and transolver++ and implement these experiments on Darcy flow and Airfoil.
>
> | Equation |            ours             |      LaMO        |  transolver++         |
> |------------|-----------------------|-----------------------|---------------------|
> | Darcy  | $ 0.0021$ | $0.0038$  | $0.0087$|
> | Airfoil     | $ 0.0020 $ | $0.0035$   | $0.0048$|
>
> We find out that, in general, using explicit AFD+preprocessing+standard neural operators has a stronger performance than using standard neural operators standalone. However, explicit AFD+preprocessing+standard neural operators still underperform AFMO, since they do not have a clear and explicit connection to the AFD theory.

---

> > ### Author Response · Authors · 2025-11-25
> > **Rebuttal to Reviewer ceLB's comments (Part 4)**
> >
> > ### You claim LaMO has low-pass filtering bias, but doesn't your method also have similar frequency characteristics since its also has an SSM transfer function, which is also a rational function.
> >
> > We appreciate the reviewer for the comment. We want to clarify that, while both use rational functions, LaMO typically relies on a fixed discretization of a predefined state matrix structure, which imposes specific constraints on the pole locations, often biasing them toward low frequencies. AFMO, via the TM system, learns the poles directly in the complex plane without being tied to a fixed discretization scheme of a linear ODE. This allows AFMO to select poles freely near the unit circle to capture high-frequency and singular features which LaMO smooths out, as showcased by the phase error analysis in Figure 3.

---

### Author Response · Authors · 2025-12-03
**Summary of rebuttal discussions**

We thank all the reviewers for conducting a thorough review. We adequately addressed all questions and comments and updated our manuscript, and the scores were raised from 6,6,4,2 to 6,6,6,4 (before emergency reversion). To summarize our work and rebuttal discussions:

## Key contributions:

AFMO introduces the first principled integration of Adaptive Fourier Decomposition (AFD) theory with frequency-domain Mamba state-space models (SSMs) to build an interpretable and scalable neural operator for PDE learning. AFMO outperforms FNO/LaMO/Transolver across structured grids, irregular domains, and singular PDE regimes.

During rebuttal, we clarified the novelties of AFMO:

- **Theory-guided design**: All components, including lifting operator, TM layer, bidirectional SSM, and aggregation, are derived directly from AFD steps rather than chosen heuristically. This yields an interpretable operator framework with modal explanations and convergence guarantees.

- **Transforming classic AFD into a learnable neural layer**: The classic AFD theory is iterative, signal-dependent, and unsuitable for deep learning. AFMO transforms AFD into a differentiable learnable layer by predicting optimal poles via an MLP, eliminating explicit greedy search and enabling end-to-end training.

- **TM-restricted transfer functions**: Unlike general rational transfer function SSMs (e.g., LaMO), AFMO constrains SSM kernels to Takenaka-Malmquist (TM) forms so that basis functions are orthonormal in RKHS. This ensures mathematical validity, stability, and resolution invariance on irregular geometries, properties not guaranteed by generic SSM operators.

## Summary of clarifications and Q&A

### Expanding baseline methods

We added BENO, UNO, Transolver++ and LinearNO on Darcy and Navier-Stokes benchmark problems. We also added Laplace Neural Operator (LNO) comparison for regular-grid PDEs. Across all tested problems, **AFMO achieved lower relative errors than these recent SOTA models**.

### Validation on large-scale 3-D problems

To address concerns about scalability and applicability to 3-D problems, we added *new 3D Brusselator PDE experiments* using the LNO dataset. **AFMO outperforms LNO on train/val/test**, demonstrating generalization to high-dimensional spatial domains and validating the scalability of the TM-SSM architecture beyond 2-D cases.

### Resolution invariance tests

 We trained AFMO, FNO and LaMO solvers at 64×64 on 2-D Darcy flow problem and conducted a zero-shot test on the performance of these solvers at 128×128 and 256×256. We observed that **AFMO maintained the lowest error growth, outperforming both LaMO and FNO**. LaMO exhibited noticeable low-pass bias, while AFMO avoided this using freely learned adaptive poles near the unit circle, preserving high-frequency details.

### Evaluation using real-world noisy data

To address concerns that experiments were conducted on synthetic datasets, we report a new experiment using the **latex glove Digital Image Correlation (DIC) dataset**, where AFMO predicts mechanical displacement fields directly from sensor data. Compared with *IFNO (state-of-the-art), FNO, and physics-based Mooney-Rivlin models*, **AFMO consistently achieved the lowest prediction errors across settings**, demonstrating effectiveness in noisy, real experimental environments.

### Performance subject to parameter shifts, or out-of-distribution geometries

We conducted additional experiments on out-of-distribution (OOD) manifold distortions and OOD PDE parameters for the Navier-Stokes equation. **AFMO outperforms the recent SOTA LaMO** on the OOD experiments.

 ### Architectural transparency and interpretability

 We provided the full architectural details of the small MLP used for pole prediction. In the revised manuscript, we added visualizations of pole distribution for Darcy (elliptic PDE) and Brusselator (reaction-diffusion) problems. To help readers understand this concept, we also offered an intuitive interpretation of poles and the selection process, which will be included in the revised manuscript.

 ### Theoretical clarifications

We clarified that 1) PDE solutions lie in the closure of the TM basis in Hardy space, **validating convergence assumptions even when solutions do not lie exactly inside the model subspace**, 2) **Theorems C.6 and C.10 rigorously bound practical approximation error**, and 3) **RKHS assumptions are directly enforced** by construction through the use of the Szegő kernel within the TM layer, verifying that the theory applies concretely to the implemented architecture.

### Computational Analysis

We refined complexity analysis to show **linear scaling with grid size** since latent token counts remain small. TM basis construction overhead is negligible compared to SSM spectral transforms. Empirically, *AFMO trains 46–51% faster than LaMO with ~2.5× lower memory usage, while maintaining near-constant memory across increasing resolutions*.

---

### Meta-Review · Area_Chair_QsPP · 2026-01-23

**Summary:**

The reviewers generally indicate that:
1. The problem is well motivated as being able to solve PDEs in complex domains which require irregular meshes is a particularly important engineering problem.
2. The work theoretically well-grounded from the design of the architecture to the convergence theorems; reviewers do note that, however, some of the assumptions in the results may to too strong (or hard to verify) for many PDE systems.
3. The benchmark comparisons are not strong enough and miss several recent works that address a similar problem.
4. An extensive number of problems are considered and significant improvement over all tested methods is shown. However, the method is not scaled up to 3D and high-resolution problems which are the most practically useful.
5. The novelty of the method is not entirely clear as several pieces of the method have been proposed in previous works and the use of SSMs to learn solution operators of PDEs has been studied by several works.

**Reviewer Concerns:**

The authors have largely addressed most of the reviewer concerns. They have clarified the assumptions in their theorems and have shown that their convergence results hold generally even when truth lies outside the model space (true for most practical examples). Furthermore, they have added a 3D PDE example and have added a resolution study on the Darcy problem. While checking 1-shot super resolution is interesting, resolution invariance is verified when training on different resolutions and obtaining the same or better error with the same model; I'd suggest the author still carry out such an experiment. Furthermore, the 3D experiment is not particularly convincing as the problem is relatively simple. I highly suggest trying something like DriverML to really test the scalability of the method. Nevertheless, I think method and theory are solid and the paper is a good contribution.

**Reviewer Scores:**

Two of the reviewer have raised their score, one to a 6 and one to 4. This is inline with the strong rebuttal provided by the authors.

---

### Decision · Program_Chairs · 2026-01-26

Accept (Poster)